# MULTI-OBJECTIVE ANTIBODY DESIGN WITH CONSTRAINED PREFERENCE OPTIMIZATION

**Milong Ren**[3,4]**, Zaikai He**[1,3,4]**, Haicang Zhang**[1,2*]
[1]Medicinal Bioinformatics Center, Shanghai Jiao Tong University School of Medicine
[2]Central China Research Institute of Artificial Intelligence
[3]Institute of Computing Technology, Chinese Academy of Sciences
[4]University of Chinese Academy of Sciences

## ABSTRACT

Antibody design is crucial for developing therapies against diseases such as cancer and viral infections. Recent deep generative models have significantly advanced computational antibody design, particularly in enhancing binding affinity to target antigens. However, beyond binding affinity, antibodies should exhibit other favorable biophysical properties such as non-antigen binding specificity and low self-association, which are important for antibody developability and clinical safety. To address this challenge, we propose AbNovo, a framework that leverages constrained preference optimization for multi-objective antibody design. First, we pre-train an antigen-conditioned generative model for antibody structure and sequence co-design. Then, we fine-tune the model using binding affinity as a reward while enforcing explicit constraints on other biophysical properties. Specifically, we model the physical binding energy with continuous rewards rather than pairwise preferences and explore a primal-and-dual approach for constrained optimization. Additionally, we incorporate a structure-aware protein language model to mitigate the issue of limited training data. Evaluated on independent test sets, AbNovo outperforms existing methods in metrics of binding affinity such as Rosetta binding energy and evolutionary plausibility, as well as in metrics for other biophysical properties like stability and specificity.

## 1 INTRODUCTION

Antibodies are vital immune proteins that bind to target antigens and trigger the adaptive immune response. Antibody design is critical in developing therapeutic drugs for a wide range of diseases such as cancer, autoimmune deficiency, and virus infections, with over a hundred antibody drugs currently approved (Kaplon et al., 2022). Structurally, antibodies consist of conserved framework regions and highly variable complementarity-determining regions (CDRs). The framework regions maintain the overall antibody structure, whereas CDRs exhibit significant variability in both structure and sequence and mainly determine antigen binding. Thus, the primary objective of computational antibody design is to design CDRs that bind the target antigens and possess favorable biochemical properties.

Recent deep generative models have achieved remarkable success in computational antibody design, particularly in enhancing antigen-specific binding affinity (Luo et al., 2022; Zhu et al., 2024; Martinkus et al., 2024; Lin et al., 2024; Zhou et al., 2024). For example, DiffAb (Luo et al., 2022) employs a denoising diffusion probabilistic model for antibody structures and sequences co-design. AbX (Zhu et al., 2024) utilizes a score-based diffusion model and incorporates geometric, physical, and evolutionary constraints to guide the design process. Notably, the recent method ABDPO (Zhou et al., 2024) integrates physical energy as guidance for binding affinity within the framework of direct preference optimization.

While binding affinity is crucial, antibodies should exhibit other favorable biophysical properties such as high target specificity and low self-association, which is important for downstream developability and clinical safety. For example, non-specific (off-target) binding can potentially trigger

---

*Correspondence should be addressed to H. Zhang (zhanghaicang@sjtu.edu.cn)

unintended immune responses, causing inflammation or other syndromes (Nicholson et al., 1991; Ferrigno, 2016; Makowski et al., 2022). High self-association, mainly due to a large amount of charged aminos on CDRs, leads to antibody aggregation and results in decreased efficacy (Sule et al., 2011; Makowski et al., 2024). In wet-lab experiments, a common strategy is to generate a diverse set of candidates and then filter them based on desired properties (Watson et al., 2023; Bennett et al., 2023). However, this post-filtering approach is inefficient and has a lower success rate in designing antibodies that satisfy all specified constraints.

To address these challenges and bridge the gap between *in silico* design and real-world applications, we propose AbNovo, a framework that leverages constrained preference optimization for multi-objective antibody design. We first pre-train a score-based diffusion model for antibody structure and sequence co-design. We then employ constrained preference optimization to fine-tune the model using metrics of binding affinity—such as Rosetta binding energy and evolutionary plausibility—as objectives, while incorporating biophysical properties related to non-specific binding, self-association, and stability as constraints. During training, we utilize a primal-dual approach to iteratively optimize the policy model. In contrast to the previous framework of constrained direct preference that leverages pairwise preferences, we model the physical binding energy as continuous rewards and collect both reward and constraint values offline. We introduce additional techniques to improve training stability and performance, such as incorporating a structure-aware protein language model to alleviate the issue of scarcity of training data.

Our contributions are summarized as follows:

- We propose AbNovo, a deep generative model for multi-objective antibody design that incorporates explicit constraints representing biophysical properties critical for real-world antibody development.
- We extend the recent framework of constrained preference optimization from language model alignment to diffusion-based generative models. This includes developing a corresponding training algorithm, supported by theoretical derivations and analysis.
- We introduce additional simple yet effective techniques to enhance performance. For instance, we incorporate a structure-aware protein language model trained on massive structural data beyond antibodies to alleviate overfitting issues due to the scarcity of antibody-antigen training data.
- Experiment results show that AbNovo achieves state-of-the-art performance in metrics of binding affinity while also satisfying other biophysical properties.

## 2 RELATED WORKS

### 2.1 COMPUTATIONAL ANTIBODY DESIGN

Antibody design is to optimize the antibody structure and sequences, particularly the CDRs, to bind to target antigens while meeting other biophysical properties. Traditional methods rely on computationally intensive Monte Carlo-based simulations to optimize the antibody-antigen binding energy (Adolf-Bryfogle et al., 2018).

Recent deep learning-based methods for antibody design can be broadly categorized into discriminative models and generative models. Discriminative models typically leverage graph neural networks to learn the presentations of antigen structure and to predict the most likely antibody structure and sequence (Kong et al., 2023; Lin et al., 2024). In contrast, generative models such as the denoising diffusion probabilistic models (DDPM) (Luo et al., 2022; Martinkus et al., 2024; Tan et al., 2024) and score-based diffusion models (Zhu et al., 2024; Kulytė et al., 2024) build an antigen-conditioned generative process of antibody sequences and structure.

Another trend is incorporating guidance into the generative process. AbX(Zhu et al., 2024) leverages evolutionary, physical, and geometric constraints to narrow down the plausible structure and sequence sampling space. The method most related to our work is AbDPO (Zhou et al., 2024), which applies preference optimization to enhance binding affinity. Our method differs from Ab-DPO in terms of framework and training objective. First, AbDPO focuses on optimizing towards a lower Rosetta energy using the framework of direct preference optimization (DPO), whereas our method employs constrained preference optimization that optimizes binding affinity while imposing

constraints related to specificity and self-association. Second, regarding the preference objective, AbDPO uses pairwise preferences, whereas our method employs continuous rewards to model the physical binding energy.

## 2.2 PREFERENCE OPTIMIZATION OF GENERATIVE MODELS

In natural language processing, large language models (LLMs) have achieved remarkable progress in natural language generation (Achiam et al., 2023; Touvron et al., 2023; Dubey et al., 2024). To better align these models with human values and preferences, various preference optimization frameworks (Azar et al., 2024; Chen et al., 2024b; Ethayarajh et al., 2024; Rafailov et al., 2024b; Meng et al., 2024; Zeng et al., 2024) have been developed. For example, Reinforcement Learning from Human Feedback (RLHF) pre-trains an explicit reward model and then fine-tunes the base models through reinforcement learning (Ouyang et al., 2022). Direct Preference Optimization (DPO) (Rafailov et al., 2024b) derives a closed form for the optimal policy and directly fine-tunes the base model with preferences data rather than an explicit reward model. More recently, DPO has been adapted from LLMs alignment to the diffusion-based generative models for image generation (Wallace et al., 2024).

However, RLHF presents several challenges in practice. First, the overoptimization issue arises (Gao et al., 2023; Coste et al., 2023; Eisenstein et al., 2023), as the reward models often serve as an imperfect proxy for true human preferences. Second, a single reward with scalar output is often inadequate to capture multiple aspects of human preferences, such as helpfulness and harmlessness, which are not always easily compatible (Ouyang et al., 2022; Ganguli et al., 2022; Thoppilan et al., 2022). To mitigate these issues, constrained RLHF (Moskovitz et al., 2023) has been proposed, which fine-tunes LLMs by maximizing a target reward while imposing explicit constraints on auxiliary safety objectives like harmlessness. More recently, the framework of constrained DPO (Liu et al., 2024; Huang et al., 2024) has also been proposed for LLMs alignment.

Our method is closely related to constrained DPO, but we extend it from LLMs alignments to the diffusion-based generative models, providing rigorous theoretical derivation. Furthermore, by integrating noise contrastive estimation with constrained DPO, we model physical binding energy using continuous rewards instead of relying on pairwise preference data. Specifically, we evaluate both reward and constraint values for sampled antibodies using existing models that assess antibody biophysical properties.

## 3 METHODS

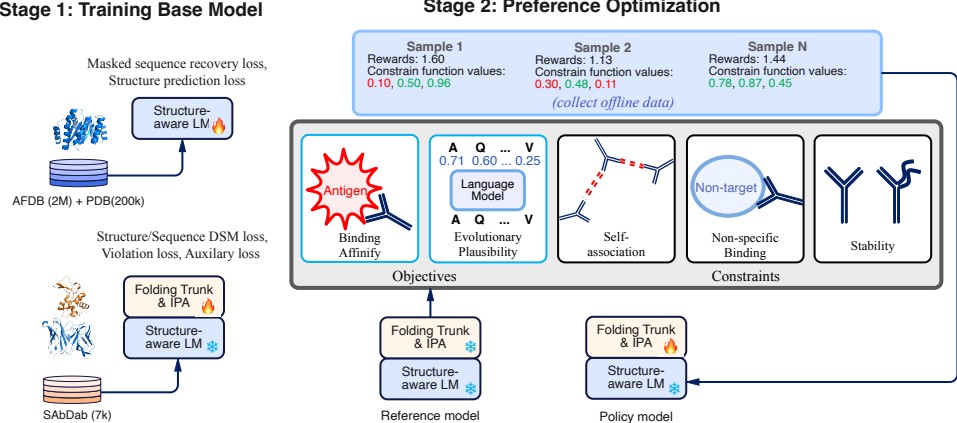

Figure 1: The overview of AbNovo. In the first stage, we train the base model, a diffusion-based generative model for the co-design of antibody CDR sequences and structures. In the second stage, we update the policy model guided by biophysical properties, using a reference model initialized from the base model as a regularization term during this update.

As illustrated in Figure 1, AbNovo comprises two main phases during training. First, we train an antigen-conditioned generative model for the co-design of antibody structure and sequence. We refer to this co-design model as the base model. Next, we fine-tune the base model using constrained preference optimization. In this process, we refer to the network being optimized as the policy model. The policy model is optimized based on biophysical properties such as Rosetta Binding Energy and self-association, while a reference model is used as a regularization term during this process, preventing over-optimization. The reference model is initialized from the base model. We briefly introduce the preliminaries and notations used in our methods in Section 3.2. Subsequently, we describe the base model and the framework of constrained preference optimization in Sections 3.2 and 3.3, respectively.

## 3.1 PRELIMINARIES

Each antibody consists of two identical heavy (H) chains and two identical light (L) chains. The variable domain of each chain is divided into framework regions and three complementarity-determining regions (CDRs): H1, H2, and H3 in the heavy chain, and L1, L2, and L3 in the light chain.

Following previous works (Luo et al., 2022; Campbell et al., 2024; Zhu et al., 2024), the structure is represented as elements of SE(3) to capture the local *frame* along the backbone (Jumper et al., 2021). For antigen-antibody complexes of total length $N$, each residues can be represented as $T_i = (x_i, r_i, a_i)$, where $i = 1, \ldots, N$. Here, $x_i \in \mathbb{R}^3$ is the coordinate of the $C_\alpha$ of the $i$-th residue, $r_i \in \mathrm{SO}(3)$ is the rotation matrix of local *frame* respect to global *frame*, and $a_i \in \{1, 2, \ldots, 20\} \cup \{[\mathrm{mask}]\}$ is one of 20 types residues or the mask state $[\mathrm{mask}]$. We assume that all CDRs to be generated has $m$ residues in total, which can be represented by $\mathcal{P}_{\mathrm{CDR}} = \{(x_i, r_i, a_i) | i = n + 1, \ldots, n + m\}$. The antibody framework and antigen can be represented by $\mathcal{P}_{\mathrm{FC}} = \{(x_i, r_i, a_i) | i = \{1, 2, \ldots, N\} \setminus \{n + 1, \ldots, n + m\}\}$. Formally, our goal is to model the distribution of $\mathcal{P}_{\mathrm{CDR}}$ conditioned on $\mathcal{P}_{\mathrm{FC}}$. We provide all the notions and their corresponding descriptions in Table 4.

## 3.2 ANTIBODY STRUCTURE AND SEQUENCE CO-DESIGN

**Diffusion Process for Sequence and Structure.** Following previous works (Yim et al., 2023; Campbell et al., 2024; Zhu et al., 2024), we use a multi-modal diffusion model to *jointly* design antibody sequence and structure. Specifically, we use the Continuous Time Markov Chain (CTMC)-based diffusion model for discrete sequence and score-based SE(3) diffusion model for structure, respectively.

We use $\mathbf{T}^{(t)} = \{T_n^{(t)}\}_{i=n+1}^{n+m} = \{(x_i^{(t)}, r_i^{(t)}, a_i^{(t)})\}_{i=n+1}^{n+m}$ to represent the antibody's structure and sequence at time $t$ and $\mathbf{T}^{(0:1)} = (\mathbf{T}^{(0)}, \mathbf{T}^{(\Delta t)}, ..., \mathbf{T}^{(1)})$ to represent the diffusion path. Here, $t$ follows a uniform distribution $\mathcal{U}(0, 1)$ and we apply distinct noise schedules on $t$ for translation $x_i^{(t)}$, rotation $r_i^{(t)}$, and sequence $a_i^{(t)}$, as described in Appendix A.4.2. The forward diffusion for $(x_i^{(t)}, r_i^{(t)}, a_i^{(t)})$ can be written as following:

$$
\begin{aligned}
x_i^{(t)} &\sim \mathcal{N}\Big(x_i^{(0)} e^{-t/2}, (1 - e^{-t})\mathrm{Id}_3\Big), \\
r_i^{(t)} &\sim \mathcal{IG}_{\mathrm{SO}(3)}\Big(r_i^{(0)}, t\Big), \\
a_i^{(t)} &\sim \mathrm{Cat}\Big(\delta\{a_i^{(0)}, a_i^{(t)}\}(1 - t) + \delta\{a_i^{(1)}, a_i^{(t)}\}t\Big).
\end{aligned}
\tag{1}
$$

Here, $\delta\{i, j\}$ is the Kronecker delta function which is 1 when $i$ equals to $j$ and is 0 otherwise. $\mathcal{N}, \mathcal{IG}_{\mathrm{SO}(3)}$, and $\mathrm{Cat}$ represent the Gaussian, Isotropic Gaussian, and Categorical distributions, respectively. We use $s_\theta^x$, $s_\theta^r$, and $s_\theta^a$ to represent the score network of translation, rotation, and sequence, respectively. We select the prior distribution of structure and sequence as $p_{\mathrm{prior}}^{x,r}(x_i^{(1)}, r_i^{(1)}) = \big(P_\#(\mathcal{N}(0, \mathrm{Id}_3)^{\otimes N}) \otimes \mathcal{IG}_{\mathrm{SO}(3)}(0, \mathrm{Id})^{\otimes N}\big)$ and $p_{\mathrm{prior}}^a(a_i^{(1)}) = \{[\mathrm{mask}]\}^{\otimes N}$, respectively. Then, the reverse diffusion process can be written as follows:

$$
\begin{aligned}
x_i^{(t-\Delta t)} &\sim P_\#\Big(\mathcal{N}\Big(\frac{1}{2}\Delta t \cdot x_i^{(t)} + \Delta t \cdot s_\theta^x(x_i^{(t)}, \mathbf{T}^{(t)}, \mathcal{P}_{\mathrm{FC}}, t), \Delta t \cdot \mathrm{Id}_3\Big)\Big), \\
r_i^{(t-\Delta t)} &\sim \mathcal{IG}_{\mathrm{SO}(3)}\Big(\exp_{r_i^{(t)}}(\Delta t \cdot s_\theta^r(r_i^{(t)}, \mathbf{T}^{(t)}, \mathcal{P}_{\mathrm{FC}}, t)), \Delta t \cdot \mathrm{Id}\Big), \\
a_i^{(t-\Delta t)} &\sim \mathrm{Cat}\Big(\delta\{a_i^{(t)}, a_i^{(t-\Delta t)}\} + \Delta t \cdot s_\theta^a(a_i^{(t-\Delta t)}, \mathbf{T}^{(t)}, \mathcal{P}_{\mathrm{FC}}, t)\Big),
\end{aligned}
\tag{2}
$$

where $\exp$ and $\log$ are the exponential and logarithmic maps, and $P_\#$ is the projection matrix that removes the center of mass $\frac{1}{N}\sum_{i\in\mathcal{P}_{\mathrm{CDR}}} x_i$.

**Network Architecture.** Inspired by previous works (Huguet et al., 2024; Zhu et al., 2024; Ren et al., 2024), we utilize a score network similar to the architecture proven useful in protein structure prediction (Lin et al., 2023; Jumper et al., 2021). It comprises three components: a structure-aware language model we pre-trained on 2.2M structures, the Evoformer trunks, and an Invariant Point Attention (IPA) (Jumper et al., 2021). We first extract embeddings from the structure-aware language model for both the conditional sequences of antibody framework and antigen, and noisy CDRs sequence. Structural information is encoded using a distogram of the antibody framework, antigen, and noisy CDRs, followed by a linear projection. Then, these structure and sequence representations are then processed by Evoformer trunks. Finally, the IPA and an MLP output the denoised CDRs structure and sequence, respectively. Appendix 12 and Figure 4 present more details on feature preprocessing and network architecture.

**Training Losses.** Following previous methods (Campbell et al., 2024; Zhu et al., 2024), the losses for training the base model primarily include Denoised Score Matching (DSM) losses for structure and sequence, auxiliary losses, and structural violation loss. Details of these losses are provided in Appendix A.4.5 presents the details of these losses.

## 3.3 Constrained Preference Optimization

To design antibodies that bind to target antigens while satisfying other biophysical properties, the training objective is formulated as:

$$
\max_\theta \; \mathcal{J}^{(\mathcal{R})}(p_\theta) = \mathbb{E}_{\mathcal{P}_{\mathrm{FC}}\in\mathcal{D}}\Big[\mathbb{E}_{\mathbf{T}^{(0)}\sim p_\theta^{x,r,a}(\mathbf{T}^{(0)}|\mathcal{P}_{\mathrm{FC}})}\Big[\sum_m \omega_m \mathcal{R}_m(\mathbf{T}^{(0)},\mathcal{P}_{\mathrm{FC}})\Big]
$$
$$
- \beta\mathbb{D}_{\mathrm{KL}}\Big(p_\theta^{x,r,a}(\mathbf{T}^{(0:1)}|\mathcal{P}_{\mathrm{FC}})\|p_{\mathrm{ref}}^{x,r,a}(\mathbf{T}^{(0:1)}|\mathcal{P}_{\mathrm{FC}})\Big)\Big], \tag{3}
$$
$$
\text{s.t.} \;\; \mathbb{E}_{\mathcal{P}_{\mathrm{FC}}\in\mathcal{D},\mathbf{T}^{(0)}\sim p_\theta^{x,r,a}(\mathbf{T}^{(0)}|\mathcal{P}_{\mathrm{FC}})}\Big[\mathcal{C}_n(\mathbf{T}^{(0)},\mathcal{P}_{\mathrm{FC}})\Big] < C_n, \;\; \text{for all } n \text{ in constraints sets,}
$$

where $\mathcal{R}_m$ and $\omega_m$ denote a normalized reward (e.g., Rosetta Binding Energy, Evolutionary Plausibility) and its weight, and $\mathcal{C}_n$ and $C_n$ denote a constraint (e.g., Non-specific Binding, Self-association, Stability) and its threshold. In this equation, the first term of the objective is to maximize rewards, while the second term is to keep the policy model close to the reference model. Additionally, the optimization is constrained to ensure that expected constraints do not exceed their thresholds. Both the reward and constraint values of sampled antibodies are computed offline, as detailed in Appendix A.5.

This problem can be associated with a Lagrangian function and then the max-min optimization problem follows:

$$
\max_{p_\theta}\min_{\boldsymbol{\lambda}\in\mathbb{R}_+^N}\mathcal{J}^{(L)}(p_\theta,\boldsymbol{\lambda}) = \mathcal{J}^{(\mathcal{R})}(p_\theta) - \mathcal{J}^{(\mathcal{C})}(p_\theta,\boldsymbol{\lambda}), \tag{4}
$$

$$
\mathcal{J}^{(\mathcal{C})}(p_\theta,\boldsymbol{\lambda}) = \sum_n \lambda_n\Big[\mathbb{E}_{\mathcal{P}_{\mathrm{FC}}\in\mathcal{D},\mathbf{T}^{(0)}\sim p_\theta^{x,r,a}(\mathbf{T}^{(0)}|\mathcal{P}_{\mathrm{FC}})}\Big(\mathcal{C}_n(\mathbf{T}^{(0)},\mathcal{P}_{\mathrm{FC}}) - C_n\Big)\Big], \tag{5}
$$

where $\mathcal{J}^{(\mathcal{R})}$ is the objective function in Equation 3 and $\boldsymbol{\lambda} = [\lambda_1,...,\lambda_n]$ is the vector of dual variables. This equation can be interpreted as appending a penalty $\mathcal{J}^{(\mathcal{C})}$ to the original objective $\mathcal{J}^{(\mathcal{R})}$. The penalty term, which depends on how much the antibody generated by the current model violates constraints, can be adjusted dynamically through the Lagrange multipliers $\boldsymbol{\lambda}$. Then this problem can be solved using the primary-dual method (Bertsekas, 2014; Ito & Kunisch, 2008; Liu et al., 2024) by iteratively taking two steps :

1. **Update policy**: Update network parameters $\theta$ to find $p_{\boldsymbol{\lambda}}^* = \arg\max_{p_\theta}\mathcal{J}^{(L)}(p_\theta,\boldsymbol{\lambda})$ based on current value of $\boldsymbol{\lambda}$.

2. **Update $\boldsymbol{\lambda}$**: Update $\boldsymbol{\lambda}$ by estimating the gradient of dual function $\mathcal{G}(\boldsymbol{\lambda}) = \mathcal{J}^{(L)}(p_{\boldsymbol{\lambda}}^*,\boldsymbol{\lambda})$ based on policy $p_{\boldsymbol{\lambda}}^*$.

Prior work has proved that the objective function is concave over $p_\theta$, and thus strong duality holds (Liu et al., 2024; Huang et al., 2024). A formal proof is provided in the Appendix A.3.1.

More specifically, we first sample a set of antibodies, denoted as $\mathcal{D}_g$, from the reference model $p_{\text{ref}}$ and compute all reward and constraint values offline for these samples. Then, based on $\mathcal{D}_g$, we iteratively perform the two steps to optimize the policy model $p_\theta$, as detailed in sections 3.3.1 and 3.3.2, respectively.

### 3.3.1 UPDATE POLICY

Given the reference model and current value of $\lambda$ along with corresponding preference data, previous methods (Rafailov et al., 2024b; Liu et al., 2024; Huang et al., 2024) leverage the DPO framework to determine $p_\lambda^*$. Here, we have significantly adapted this framework for the antibody design task.

**Direct Perference Optimization with Continuous Rewards.** Recent work of NCA (Chen et al., 2024a) has extended the DPO framework to incorporate continuous reward values for large language model alignment. Since many biophysical properties (e.g., physical energy) are continuous values, we have further adapted NCA for diffusion-based generative models and integrated it into the constrained preference optimization framework. The detailed theoretical derivation is presented in the Appendix A.3.2. Then, the optimal policy $p_\lambda^*$ has the following form:

$$p_\lambda^* \propto p_{\text{ref}} \exp(\hat{\mathcal{R}}/\beta), \tag{6}$$

$$\hat{\mathcal{R}} = \sum_m \omega_m \mathcal{R}_m(\mathbf{T}^{(0)}, \mathcal{P}_{\text{FC}}) - \sum_n \lambda_n \mathcal{C}_n(\mathbf{T}^{(0)}, \mathcal{P}_{\text{FC}}), \tag{7}$$

where $\mathbf{T}^{(0)} \in \mathcal{D}_g$ are the sampled antibodies under the reference model, and $\beta$ is the regulartizion weight in Equation 3.

Integrating with NCA, the original training objective in Equation 4 can be reformulated as follows (Appendix A.3.2):

$$\mathcal{L}_{\text{NCA}}^{\text{diff}}(\theta) = -\mathbb{E}_{\{\mathbf{T}_i^{(0)}, \mathcal{P}_{\text{FC}}, \hat{\mathcal{R}}_i\}_{1:K} \in \mathcal{D}_g} \sum_{i=1}^{K} \left[ \frac{\exp\left(\hat{\mathcal{R}}_i/\beta\right)}{\sum_{j=1}^{K} \exp\left(\hat{\mathcal{R}}_j/\beta\right)} \log \sigma\left(f_i\right) + \frac{1}{K} \log \sigma\left(-f_i\right) \right],$$

$$f_i = \mathbb{E}_{\mathbf{T}^{(\Delta t:1)} \sim p_\theta^{x,r,a}(\mathbf{T}^{(\Delta t:1)}|\mathbf{T}_i^{(0)}, \mathcal{P}_{\text{FC}})} \left[ \log \frac{p_\theta^{x,r,a}(\mathbf{T}^{(0:1)}|\mathcal{P}_{\text{FC}})}{p_{\text{ref}}^{x,r,a}(\mathbf{T}^{(0:1)}|\mathcal{P}_{\text{FC}})} \right]. \tag{8}$$

Here, we use $\mathcal{D}_g$ to represent antibodies sampled from the reference model and $\sigma$ is the sigmoid function. In this equation, the first term is to increase the likelihood of the samples based on their rewards and the second one serves as a regularization term.

Due to the objective in Equation 8 is inefficient and intractable to train, we utilize Jensen's inequality and approximate the reverse process with the forward diffusion $q^{x,r,a}$ (Wallace et al., 2024; Zhou et al., 2024). As derived in Appendix A.3.2, the simplified objective is as follows:

$$\mathcal{L}_{\text{NCA}}^{\text{diff}}(\theta) = -\mathbb{E}_{\substack{t \sim \mathcal{U}(0,1), \{\mathbf{T}_i^{(0)}, \mathcal{P}_{\text{FC}}, \hat{\mathcal{R}}_i\}_{1:K} \in \mathcal{D}_g, \\ \mathbf{T}_i^{(t)} \sim q^{x,r,a}(\mathbf{T}_i^{(t)}|\mathbf{T}_i^{(0)})}} \sum_{i=1}^{K} \left[ \frac{\exp\left(\hat{\mathcal{R}}_i/\beta\right)}{\sum_{j=1}^{K} \exp\left(\hat{\mathcal{R}}_i/\beta\right)} \log \sigma\left(\frac{1}{\Delta t}\mathcal{F}_i\right) + \frac{1}{K} \log \sigma\left(-\frac{1}{\Delta t}\mathcal{F}_i\right) \right],$$

$$\mathcal{F}_i = -\mathbb{D}_{\text{KL}}\left(q^{x,r,a}(\mathbf{T}_i^{(t-\Delta t)}|\mathbf{T}_i^{(0,t)}, \mathcal{P}_{\text{FC}})||p_\theta^{x,r,a}(\mathbf{T}_i^{(t-\Delta t)}|\mathbf{T}_i^{(0,t)}, \mathcal{P}_{\text{FC}})\right)$$

$$+ \mathbb{D}_{\text{KL}}\left(q^{x,r,a}(\mathbf{T}_i^{(t-\Delta t)}|\mathbf{T}_i^{(0,t)}, \mathcal{P}_{\text{FC}})||p_{\text{ref}}^{x,r,a}(\mathbf{T}_i^{(t-\Delta t)}|\mathbf{T}_i^{(0,t)}, \mathcal{P}_{\text{FC}})\right). \tag{9}$$

**Increase the Likelihood of Samples with High Rewards.** Previous studies have found that the DPO-based approach makes the likelihood of the optimal preference samples decrease (Rafailov et al., 2024a). Chen et al. (2024a) demonstrated that the NCA training objectives ensure that the likelihood of the optimal reward does not decrease. Building on this, we kept the training objectives from the base model, $\mathcal{L}_{\text{sup}}$, to further increase the likelihood of samples with higher rewards. Then

the total loss can be written as:

$$\mathcal{L}_{\text{update policy}} = \mathcal{L}_{\text{NCA}}^{\text{diff}} + \alpha^{(\text{sup})} \sum_{i=1}^{K} \max\left(0, \frac{\hat{\mathcal{R}}_i - \frac{1}{K}\sum_i \hat{\mathcal{R}}_i}{\sum_j \hat{\mathcal{R}}_j}\right) \mathcal{L}_{\text{sup},i}. \tag{10}$$

### 3.3.2 UPDATE $\lambda$

This step involves calculating the gradient of the Lagrange multipliers by assessing the extent of constraint violation in the current policy. The gradient of $\mathcal{G}(\lambda)$ can be written as follows:

$$\frac{d\mathcal{G}(\lambda)}{d\lambda} = \mathbb{E}_{\mathcal{P}_{\text{FC}} \in \mathcal{D}, \mathbf{T}^{(0)} \sim p_\lambda^*(\mathbf{T}^{(0)} | \mathcal{P}_{\text{FC}})} \Big[ \mathbf{C} - \mathcal{C}(\mathbf{T}^{(0)}, \mathcal{P}_{\text{FC}}) \Big], \tag{11}$$

where $\mathbf{C} = [C_1, ..., C_n]$ and $\mathcal{C} = [\mathcal{C}_1, ..., \mathcal{C}_n]$. In this equation, the gradient of $\mathcal{G}(\lambda)$ can be calculated by the expected degree of constraint violation in the sampled antibodies under the current policy model $p_\lambda^*$.

Considering that the optimal solution of Equation 9 under $\lambda$ can be written as $p_\lambda^* = \frac{1}{Z_\lambda} p_{\text{ref}} \exp(\hat{\mathcal{R}}/\beta)$, the specific estimation approach can be deduced into a closed-form:

$$\frac{d\mathcal{G}(\lambda)}{d\lambda} = \mathbb{E}_{\mathcal{P}_{\text{FC}} \in \mathcal{D}_g} \left[ \frac{\mathbb{E}_{\mathbf{T}^{(0)} \sim p_{\text{ref}}(\mathbf{T}^{(0)} | \mathcal{P}_{\text{FC}})} \Big( \exp(\hat{\mathcal{R}}/\beta)(\mathbf{C} - \mathcal{C}(\mathbf{T}^{(0)}, \mathcal{P}_{\text{FC}})) \Big)}{\mathbb{E}_{\mathbf{T}^{(0)} \sim p_{\text{ref}}(\mathbf{T}^{(0)} | \mathcal{P}_{\text{FC}})} \exp(\hat{\mathcal{R}}/\beta)} \right]. \tag{12}$$

By this way, we can estimate the gradient of the $\mathcal{G}(\lambda)$ in an *offline* manner. The details of the estimation process are provided in Appendix A.3.3.

### 3.3.3 ITERATIVE OPTIMIZATION

Following previous work (Dubey et al., 2024), we update the reference model with the latest trained policy model and conduct several rounds of constrained preference optimization. Specifically, in the $k$-th round, we use the trained policy model from the $(k-1)$-th round to update the reference model and collect sampled antibodies and their reward and constraint values offline. The entire training process for constrained preference optimization is detailed in Algorithm 1.

## 4 RESULTS

### 4.1 EXPERIMENTS SETUP

**Training and Testing Sets.** We trained AbNovo using antibody-antigen complex structures derived from the SAbDab database (Dunbar et al., 2014) and evaluated its performance on the RAbD test set, which is widely used for *in silico* antibody design. During testing, we simultaneously generated all six CDRs conditioned on the antigen and the antibody framework regions. Following previous work, we strictly eliminated any overlap between the training and test sets by applying a 40% sequence identity threshold on CDR-H3. More details on the preparation of the training and testing datasets are provided in Appendix A.4.4.

**Baseline Methods.** We compare AbNovo with representative methods from each category: discriminative model dyMEAN (Kong et al., 2023) and GeoAb (Lin et al., 2024); and diffusion-based generative models DiffAb (Luo et al., 2022) and AbX (Zhu et al., 2024). Since dyMEAN does not use native antibody framework structure as input, its settings differ from other methods, which may lead to an underestimation of dyMEAN's performance in the comparisons presented in our experiments. We note that AbDiffuser (Martinkus et al., 2024) and AbDPO (Zhou et al., 2024) are unavailable for benchmarking. More details on running these methods are in Appendix A.6.

**Evaluation Metrics.** We group the evaluation metrics into two categories: reference-based metrics and reference-independent metrics. The reference-based metrics assess the similarity between the designed and native antibody structures and sequences. Specifically, Amino Acid Recovery (AAR, %) measures the sequence recovery accuracy by comparing the generated sequences to the native sequences. Root Mean Square Deviation (RMSD, Å) calculates the structural deviation between $C_\alpha$ coordinates of the generated and native CDRs.

---

**Algorithm 1** Constrained Preference Optimization for Antibody Design

---

1: **Input:** Antigen-antibody complex dataset $\mathcal{D}$, base model $p_\theta^{(0)}$, preference loss $\mathcal{L}$, dual function $\mathcal{G}$, reward functions $\{\mathcal{R}_m(\mathbf{T}^{(0)}, \mathcal{P}_{\text{FC}})\}_{m=1}^M$, constraint functions $\{\mathcal{C}_n(\mathbf{T}^{(0)}, \mathcal{P}_{\text{FC}})\}_{n=1}^N$, initial vector of dual variables $\boldsymbol{\lambda}$, weights of the rewards $\{\omega_m\}_{m=1}^M$, learning rate $(\eta_\theta, \eta_\lambda)$, number of rounds of constrained preference optimization $K$, number of training steps during each round $B$, number of data for training $V$.

2: **for** $k = 1$ to $K$ **do**
    *# Initialize reference model and policy model of the current round*

3:    $p_{\text{ref}}^{(k)} \leftarrow p_\theta^{(k-1)}$

4:    $p_\theta^{(k)} \leftarrow p_\theta^{(k-1)}$
    *# Collect the antibody samples from reference model offline*

5:    $\mathcal{D}_g = \{(\mathbf{T}_i^{(0)}, \mathcal{P}_{\text{FC},i})|_{i=1}^V \text{ s.t. } \mathcal{P}_{\text{FC},i} \sim \mathcal{D}, \mathbf{T}_i^{(0)} \sim p_{\text{ref}}^{(k)}(\mathbf{T}^{(0)}|\mathcal{P}_{\text{FC},i})\}$
    *# Compute the reward and constraint values of the antibody samples in $\mathcal{D}_g$ offline*

6:    $\mathcal{R}_{m,i}, \mathcal{C}_{n,i} \leftarrow \text{Evaluate}(\mathcal{D}_g)$

7:    **for** $b = 1$ to $B$ **do**
        *# Annotate $\mathcal{D}_g$ with Equation 7 under current $\boldsymbol{\lambda}$*

8:        $\hat{\mathcal{R}}_i(\mathbf{T}_i^{(0)}, \mathcal{P}_{\text{FC}}) = \sum_m \omega_m \mathcal{R}_{m,i} - \sum_n \lambda_n \mathcal{C}_{n,i}$
        *# Update policy model $\theta$ as Equation 9*

9:        $p_\theta^{(k)} \leftarrow \text{PolicyOptimizer}(p_\theta^{(k)}, \mathcal{L}, \eta_\theta, \mathcal{D}_g)$
        *# Update $\boldsymbol{\lambda}$ as Equation 12*

10:       $\boldsymbol{\lambda} \leftarrow \text{LambdaOptimizer}(\boldsymbol{\lambda}, \mathcal{G}, \eta_\lambda, \mathcal{D}_g)$

11:    **end for**

12: **end for**

13: **return** $p_\theta^{(K)}$

---

The reference-independent metrics evaluate properties without direct comparison to native antibodies. Rosetta Binding Energy assesses the binding affinity of the designed antibodies to target antigens. Evolutionary Plausibility is evaluated using the likelihood under an independent antibody language model (Shuai et al., 2021). Additionally, we consider the proportion of generated antibodies that satisfy constraints related to self-association, stability, and non-specific binding. For each method, we designed 128 antibodies per antigen and evaluated their average metrics. Details of these metrics and the thresholds for each constraint are provided in Appendix A.5.

### 4.1.1 EVALUATION ON MULTI-OBJECTIVE ANTIBODY DESIGN

As shown in Table 1, AbNovo outperforms all baseline methods across all reference-independent metrics. These results indicate that AbNovo not only excels in designing antibodies with superior binding energy and evolutionary plausibility but also achieves the lowest percentage of constraint violations compared to other methods. Furthermore, in comparison to our base model, AbNovo demonstrates significant improvements across all metrics, underscoring the effectiveness of constrained preference optimization.

Subsequently, we evaluated AbNovo on reference-based metrics. As shown in Table 5, our method achieves superior performance on these metrics compared to all baseline methods.

### 4.2 ABLATION STUDIES

We trained several ablation models to investigate the relative importance of the core components of our method.

First, we compare constrained preference optimization with supervised fine-tuning (SFT). In the SFT setting, we selected the optimal sample for each antigen as training data, based on the weighted rewards for each property (SFT in Table 2). As presented in Table 2, while SFT improves the performance of the base model, it remains less effective compared to preference optimization learning.

---

[1]GeoAb only designs the CDR H3 of the antibody. For the other CDRs, we utilize the natural antibody sequence and structure for evaluation.

Table 1: Evaluation of reference-independent metrics on RAbD test set. Here, *reference* represents the native antibody structure and sequence in RAbD dataset.

| Methods | Binding Energy (↓) | Evolutionary Plausibility (↓) | Self-association (↓) | Stability (↓) | Non-specific Binding (↓) |
|---|---|---|---|---|---|
| *reference* | -19.41 | 2.38 | 12.3% | 0% | 3.5% |
| DiffAb | -0.96 | 2.60 | 7.6% | 15.6% | 2.3% |
| dyMEAN | -1.74 | 2.82 | 50.8% | 94.5% | 1.8% |
| GeoAb [1] | -1.75 | 2.69 | 38.5% | 7.6% | 4.3% |
| AbX | 4.79 | 2.44 | 14.5% | 4.8% | 11.6 % |
| AbNovo (base) | -2.60 | 2.41 | 19.9% | 2.9% | 10.9% |
| AbNovo | **-12.05** | **2.36** | **2.3%** | **2.8%** | **1.7%** |

Second, we compare constrained preference optimization with preference optimization used in Ab-DPO (Zhou et al., 2024). For preference optimization, we convert all constraints considered in AbNovo into optimization objectives (Multi-objective in Table 2). We observed a slight increase in fulfilling all constraints but a significant drop in performance in Binding Energy and Evolutionary Plausibility. This indicates that for antibody design, certain biophysical properties are more suitably treated as constraints rather than optimization objectives. This phenomenon has also been observed in previous studies on language model alignment (Liu et al., 2024; Huang et al., 2024).

Third, we included two ablation experiments to demonstrate the relative contribution of the structure-aware language model that is used to alleviate the scarcity of antibody-antigen complex data. *i*) We trained an ablation model where we excluded the embeddings of the language model as input features (w.o. LM in Table 2). We observed significant drops across nearly all metrics, indicating the importance of the language model. *ii*) We also trained an ablation model where we replaced this structure-aware language model with a sequence-only language model (ESM-2 based in Table 2). It shows that the structure-aware model yielded better results than the sequence-only language model. Additionally, we assessed its ability to predict long-distance contacts (Appendix A.2.4) and found that the structure-aware language model significantly outperforms the pure sequence language model on independent test sets, including the CASP and CAMEO datasets.

Finally, we analyzed the effect of varying the number of iteration rounds $K$ in constrained preference optimization on performance. As shown in Table 3, we observed that increasing the number of iteration rounds resulted in improvements across overall metrics, with a higher proportion of generated samples satisfying the constraints.

Table 2: Ablation studies for AbNovo on RAbD dataset. Ablation studies for AbNovo on the RAbD dataset. The ablation experiment settings include: without using a language model (w.o. LM), replacing the structure-aware language model with ESM2 (ESM-2 based), using supervised fine-tuning instead of preference optimization (SFT), and incorporating all constraints into the optimization objectives (Multi-objective).

| Methods | Binding Energy (↓) | Evolutionary Plausibility (↓) | All Constrains (↓) | AAR (↑) | RMSD (↓) |
|---|---|---|---|---|---|
| w.o. LM | 7.54 | 2.67 | 46.5% | 41.53% | 3.19 |
| ESM-2 based | 1.75 | 2.40 | 30.8% | 49.2% | 2.55 |
| AbNovo (base) | -2.60 | 2.41 | 26.7% | **49.9**% | **2.19** |
| SFT | -6.46 | 2.37 | 6.5% | 48.8% | 2.41 |
| Multi-objective | -4.05 | 2.39 | **2.6**% | 42.7% | 2.43 |
| AbNovo | **-12.05** | **2.35** | 3.9% | 48.5% | 2.37 |

## 4.3 CASE STUDIES

We present a case study (Figure 2) comparing the designed antibodies from different methods: dyMEAN, DiffAb, and AbNovo. This case illustrates that antibodies designed by AbNovo not

Table 3: Improvements achieved through iterative constrained preference optimization. Iter1-4 represent the model's performance under constrained preference optimization with varying numbers of iterations.

| Methods | Binding Energy ($\downarrow$) | Evolutionary Plausibility ($\downarrow$) | Self-association ($\downarrow$) | Stability ($\downarrow$) | Non-specific Binding ($\downarrow$) |
|---|---|---|---|---|---|
| AbNovo (base) | -2.60 | 2.41 | 19.9% | 2.9% | 10.9% |
| Iter1 | -6.45 | 2.38 | 6.5% | 8.9% | 4.7% |
| Iter2 | -11.60 | 2.38 | 2.0% | 5.2% | 5.6% |
| Iter3 | -12.05 | 2.36 | 2.3% | 2.8% | 1.7% |

only exhibit higher binding affinity to target antigens but also fully satisfy all constraints. Previous studies have shown that a larger area of negatively charged patches in the CDRs corresponds to a higher risk of self-association in wet-lab experiments (Makowski et al., 2024). We see that dyMEAN produce a large number of charged amino acids which can lead to potential risks of self-association.

In detail, we present the distribution of metrics for antibodies designed by different methods for specific antigens. As shown in Figure 3, for AbNovo-designed antibodies, there is not only a higher percentage of antibodies that satisfy the constraints but also a greater proportion that outperforms natural antibodies across binding energy and evolutionary plausibility.

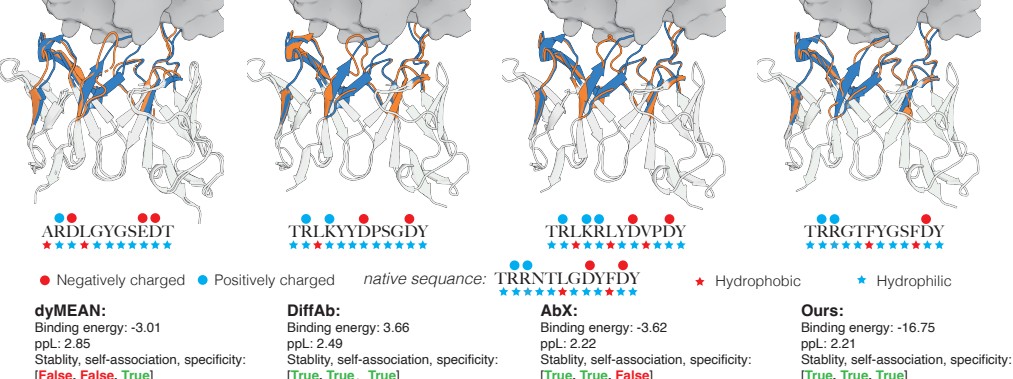

Figure 2: Visualization of the antibody designed by dyMEAN, DiffAb, and AbNovo for given antigens (PDB ID: 5NUZ). We used orange and blue to identify all CDRs of the designed antibody and all CDRs of the natural antibody. False and True are used to indicate whether the designed antibody satisfies different constraints or not. We presented the CDR H3 sequences designed by different methods, indicating the biophysical properties, such as hydrophilicity and hydrophobicity.

## 5 DISCUSSION

To design antibodies with strong affinity while enhancing their developability and clinical safety, we propose AbNovo, a constrained preference optimization framework. Experimental results demonstrate that the AbNovo framework improves the affinity and evolutionary plausibility of designed antibodies by ensuring specificity, stability, and minimizing self-association.

The limitations of AbNovo mainly lie in the following aspects. First, while AbNovo emphasizes various *in silico* metrics for evaluation, a limitation is the absence of *wet-lab* experimental validation, which will be addressed in our future work. Second, although AbNovo can design antibodies in scenarios with unknown antigen-antibody binding positions by integrating structure prediction and docking methods, the complexity of this pipeline may lead to error accumulation. Third, though metrics like Rosetta energy are widely used in evaluating antibody design, they still do not perfectly align with wet-lab experiments. Our framework is adaptable and can incorporate other physicochemical properties as rewards and constraints.

## CODE AVILIBILITY

Code for AbNovo can be found at `https://github.com/CarbonMatrixLab/AbNovo`.

## ACKNOWLEDGMENTS

We acknowledge the financial support provided by the National Natural Science Foundation of China (Grant No. 32370657) and the *Plan for Advancing Scientific Research Paradigms and Empowering Disciplinary Upgrades Through Artificial Intelligence* in Shanghai, awarded to H.Z.

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

# A    APPENDIX

## A.1    NOTATIONS

Table 4: Mathematical Symbol Explanations

| Symbol | Description |
|---|---|
| $\mathcal{P}_{\text{FC}}$ | Antibody framework and antigen |
| $\mathcal{P}_{\text{CDR}}$ | CDRs of antibody |
| $x \in \mathbb{R}^3$ | Coordinate of $C_\alpha$ atom |
| $r \in \text{SO}(3)$ | Rotation matrix |
| $a \in \{1, 2, \ldots, 20\} \cup \{[\text{mask}]\}$ | Amino acid type with mask |
| $t$ | Diffusion time |
| $\mathbf{T}^{(t)} = (x^{(t)}, r^{(t)}, a^{(t)})$ | Antibody's structure and sequence at time $t$ |
| $\mathbf{T}^{(0:1)} = (\mathbf{T}^{(0)}, \mathbf{T}^{(\Delta t)}, ..., \mathbf{T}^{(1)})$ | Diffusion path |
| $s_\theta$ | Score network |
| $p_{\text{prior}}$ | Prior distribution |
| $p_{\text{ref}}$ | Reference model |
| $p_\theta$ | Policy model |
| $p_{\boldsymbol{\lambda}}^*$ | Optimal policy model under $\boldsymbol{\lambda}$ |
| $\mathcal{U}$ | Uniform distribution |
| $\mathcal{N}$ | Gaussian distribution |
| $\mathcal{IG}_{\text{SO}(3)}$ | Isotropic gaussian distribution on SO(3) |
| Cat | Categorical distribution |
| $\mathcal{R}$ | Reward function |
| $\mathcal{C}$ | Constraint function |
| $\omega \in \mathbb{R}_+$ | Weight of reward |
| $C \in \mathbb{R}_+$ | Threshold of constraint |
| $\lambda \in \mathbb{R}_+$ | Dual variable |
| $\delta$ | Kronecker delta function |
| $\mathbb{D}_{\text{KL}}$ | Kullback-Leibler divergence |
| $P_\#$ | Projection matrix that removes the center of mass |
| $\mathcal{G}$ | Dual function |
| $\mathcal{D}$ | Antigen-antibody complex dataset |
| $\mathcal{D}_g$ | Antibodies sampled from the reference model |
| $\beta$ | Regularization weight |

## A.2    ADDITIONAL RESULTS

### A.2.1    REFERENCE-BASED METRICS EVALUATION

In this section, we evaluate AbNovo on reference-based metrics. As shown in Table 5, AbNovo outperforms all baseline methods.

### A.2.2    ANTIBODY OPTIMIZATION

In this section, we further evaluate AbNovo on antibody optimization tasks. Here, we specifically compare AbNovo with the generative model DiffAb and AbX. We follow the experimental process proposed by DiffAb (Luo et al., 2022). This process involves perturbing the CDR sequence and structure at time $t$ using forward diffusion, then denoising from time $t$ to time 0 in reverse diffusion to generate 128 antibodies for each antigen. We also follow the evaluation metrics for antibody

Table 5: Evaluation of reference-based metrics across each CDR in RAbD test dataset.

| | Method | AAR(%) ↑ | RMSD(Å)↓ | | Method | AAR(%) ↑ | RMSD(Å) ↓ |
|---|---|---|---|---|---|---|---|
| | DiffAb | 70.01 | 0.88 | | DiffAb | 61.07 | 0.85 |
| | dyMEAN | 75.71 | 1.09 | | dyMEAN | 75.55 | 1.03 |
| H1 | GeoAb | - | - | L1 | GeoAb | - | - |
| | AbX | 80.92 | 0.85 | | AbX | 80.37 | 0.80 |
| | AbNovo (base) | **85.25** | 0.66 | | AbNovo (base) | **84.34** | 0.66 |
| | AbNovo | 84.55 | **0.65** | | AbNovo | 83.50 | **0.65** |
| | DiffAb | 38.52 | 0.78 | | DiffAb | 58.58 | 0.55 |
| | dyMEAN | 68.48 | 1.11 | | dyMEAN | 83.09 | 0.66 |
| H2 | GeoAb | - | - | L2 | GeoAb | - | - |
| | AbX | 70.73 | 0.76 | | AbX | 84.53 | 0.45 |
| | AbNovo (base) | **78.56** | **0.61** | | AbNovo (base) | **88.25** | **0.32** |
| | AbNovo | 76.60 | 0.62 | | AbNovo | 88.05 | 0.35 |
| | DiffAb | 28.05 | 2.86 | | DiffAb | 47.57 | 1.39 |
| | dyMEAN | 37.50 | 3.88 | | dyMEAN | 52.11 | 1.44 |
| H3 | GeoAb | 41.19 | 2.57 | L3 | GeoAb | - | - |
| | AbX | 44.18 | 2.50 | | AbX | 65.89 | 1.21 |
| | AbNovo (base) | **49.93** | **2.19** | | AbNovo (base) | 73.88 | **0.86** |
| | AbNovo | 48.55 | 2.38 | | AbNovo | **74.45** | 0.86 |

optimization used in previous works (Luo et al., 2022; Zhu et al., 2024). We find that AbNovo out-performed other baseline methods in optimizing Rosetta Binding Energy, Evolutionary Plausibility (Table 6), the proportion of satisfying constraints (Table 7), AAR and RMSD (Table 8).

Table 6: Performance of Rosetta Binding Energy (left) and Evolutionary Plausibility (right) across different antibody optimization steps across DiffAb, AbX, and AbNovo.

| Optimization Steps | DiffAb | AbX | AbNovo |
|---|---|---|---|
| 4 | -10.45 / 2.39 | -8.80 /2.40 | **-21.02 / 2.39** |
| 8 | -8.52 / 2.41 | -2.64 / 2.43 | **-19.77 / 2.37** |
| 16 | -7.18 / 2.42 | 2.07 / 2.42 | **-12.70/ 2.37** |
| 32 | 0.23 / 2.53 | -3.05 / 2.44 | **-12.87 / 2.36** |
| 64 | 0.23 / 2.57 | 3.98 / 2.44 | **-12.87 / 2.36** |
| 100 | -0.96 / 2.60 | 4.79 / 2.44 | **-12.05 / 2.36** |

Table 7: Proportion of constraint violations across varying antibody optimization steps.

| Optimization Steps | DiffAb | AbX | AbNovo |
|---|---|---|---|
| 4 | 13.2 % | 14.0 % | **12.8** % |
| 8 | 13.9 % | 22.7 % | **7.1** % |
| 16 | 13.6 % | 22.5 % | **6.5** % |
| 32 | 15.7 % | 21.9 % | **4.2** % |
| 64 | 21.5 % | 23.0 % | **2.6** % |
| 100 | 20.8 % | 23.5 % | **3.9** % |

Table 8: AAR and RMSD across different antibody optimization steps.

| Optimization Steps | DiffAb | AbX | AbNovo |
|---|---|---|---|
| 4 | **0.88** / 1.09 | 0.80 / 0.97 | 0.85 / **0.80** |
| 8 | **0.76** / 1.59 | 0.59 / 1.51 | 0.66 / **1.34** |
| 16 | 0.48 / 1.78 | 0.49 / 1.54 | **0.51 / 1.46** |
| 32 | 0.39 / 2.05 | 0.45 / 1.88 | **0.50 / 1.66** |
| 64 | 0.30 / 2.69 | 0.45 / 2.33 | **0.48 / 2.03** |
| 100 | 0.28 / 2.86 | 0.44 / 2.50 | **0.49 / 2.38** |

### A.2.3 DESIGN ANTIBODIES FOR UNKNOWN BINDING POSE BETWEEN THE ANTIBODY AND ANTIGEN

To enable antibody design in scenarios with unknown antigen-antibody positions and unknown antibody frameworks, we adopted a previously established pipeline (Luo et al., 2022) (`https://github.com/luost26/diffab/tree/main`)). Specifically, we used a structure prediction method Chai-1 (Discovery et al., 2024) and docking software HDock (Yan et al., 2020) to predict the relative binding position of the antigen and antibody, followed by utilizing AbNovo for antibody design. As shown in Table 9, AbNovo demonstrated superiority among the comparative methods.

Table 9: Evaluation of design antibodies for unknown binding pose between the antibody and antigen.

| | AAR | RMSD | Rosetta Binding Energy | Evolutionary Plausibility | Constraints |
|---|---|---|---|---|---|
| dyMEAN | 0.37 | 3.88 | -1.7 | 2.82 | 94.5 % |
| AbX | 0.40 | 2.83 | 11.22 | **2.54** | 39.39 % |
| AbNovo | **0.44** | **2.59** | **-5.81** | **2.54** | **25.5%** |

### A.2.4 EVALUATE PRE-TRAINED LANGUAGE MODEL

To demonstrate the effectiveness of structure-aware language model, we have evaluated it on CASP14, CASP15, and CAMEO (from 2022-05-01 to 2023-05-01) independent datasets. We follow previous work (Lin et al., 2023) and choose long-range ($L > 24$) contact prediction accuracy (p@L) as the evaluation metric. The results are shown in Table 10.

We observe that structure-aware language model significantly outperforms the sequence pure language model (ESM-2 3B).

Table 10: Evaluate pre-trained language model on CASP14, CASP15, CAMEO dataset.

| Methods | CASP14 | CASP15 | CAMEO |
|---|---|---|---|
| ESM2-3B | 0.37 | 0.44 | 0.51 |
| Ours | **0.58** | **0.59** | **0.75** |

### A.2.5 CASE STUDIES

We analyze the metric distributions of antibodies designed by various methods for specific antigens. dyMEAN is not a generative model and lacks the ability to produce diverse outcomes, its results are excluded from this analysis. As illustrated in Figure 3, AbNovo has better performance in Rosetta Binding Energy, Evolutionary Plausibility, and the proportion of constraint satisfaction.

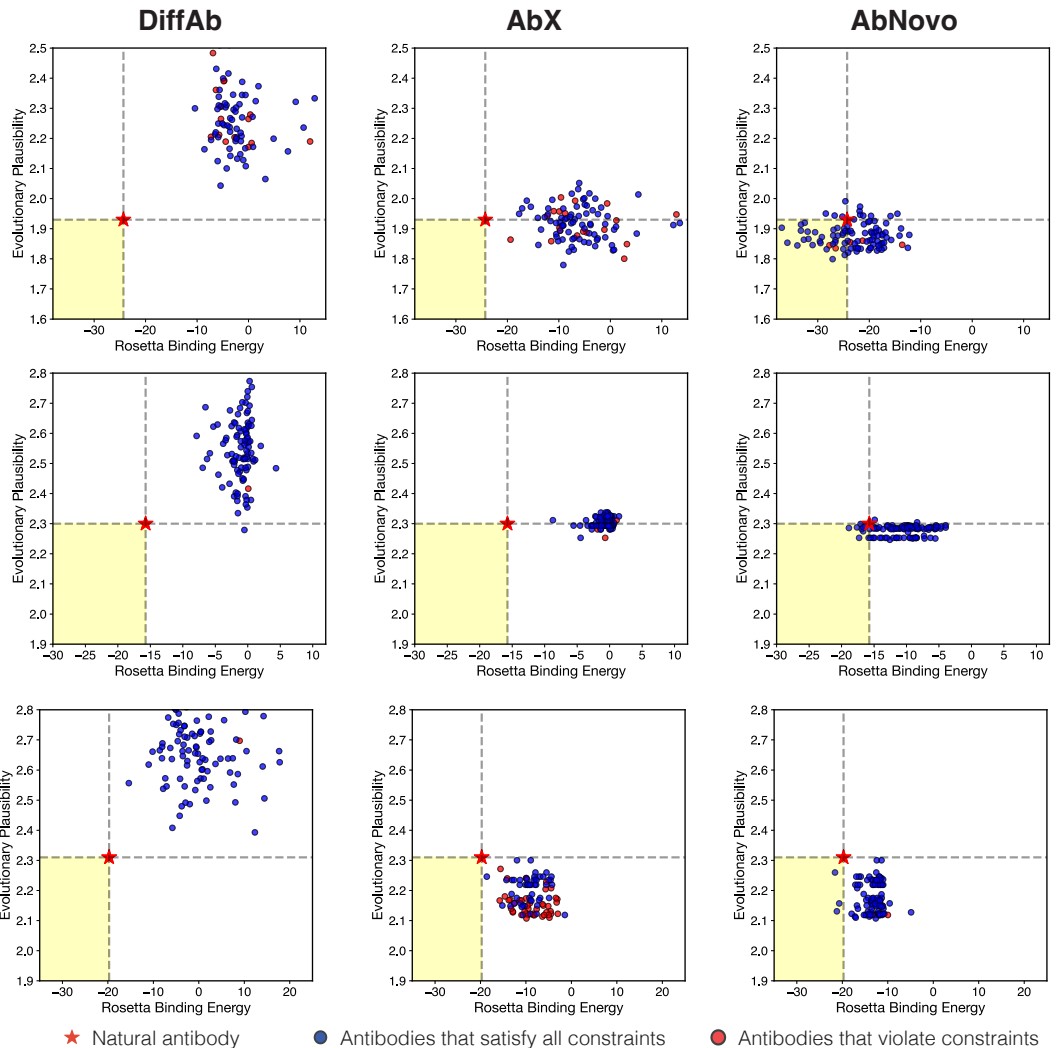

Figure 3: Distribution of Rosetta binding energy and evolutionary plausibility for 100 antibodies designed against the different antigens (PDB:4fqj, 1a2y, and 5nuz) using various methods. The red star denotes the characteristics of the natural antibodies. Antibodies that satisfy all constraints (Stability, Self-association, Non-specific Binding) are marked in blue, while those that violate the constraints are marked in red. The yellow regions highlight areas where the binding energy and evolutionary plausibility metrics exceed those of the natural antibody.

## A.3 ANALYTICAL RESULTS

### A.3.1 STRONG DUALITY OF CONSTRAINED PREFERENCE OPTIMIZATION

This section demonstrates that strong duality holds for the constrained optimization problem (Equation 3). This material is from previous works Ito & Kunisch (2008); Bertsekas (2014); Huang et al. (2024), we include it here only for completeness.

The Slater condition is an important concept in primal-dual algorithms and is mainly used to determine a strong duality of a problem. Specifically, the Slater condition requires the existence of a strictly feasible point in the constrained optimization problem, i.e., a point that satisfies all the inequality constraints and all the inequality constraints strictly hold. If Slater's condition is satisfied, it is usually possible to ensure that the optimal solution of the primal problem is equal to the optimal solution of the dual problem, thus achieving strong duality.

**Assumption 1** *(Slater condition, Strict feasible) This exists a policy $p$ and $\epsilon > 0$ such that $\mathcal{J}^{(\mathcal{C})}(p, \boldsymbol{\lambda}) < \epsilon$.*

In practice, this is possible because we often know the optimization strategy's strictly feasible solution, so we can relax the constraints appropriately according to the application scenario Liu et al. (2024). Similarly, in this manner, AbNovo can satisfy the *Slater condition*.

**Proposition 1** *(Strong duality of constrained optimization) Under Slater condition (Assumption 1), there is no duality gap for constrained optimization problem. Let $p^*$ be the optimal primal policy such that $p^* = \arg\max_{p_\theta} \mathcal{J}^{(\mathcal{C})}(p_\theta, \boldsymbol{\lambda})$. Let $\boldsymbol{\lambda}^*$ to be the optimal dual variable where $\boldsymbol{\lambda}^* = \arg\min_{\boldsymbol{\lambda} \geq 0} \mathcal{G}(\boldsymbol{\lambda})$. Then $(p^*, \boldsymbol{\lambda}^*)$ is a saddle point of the Langrangian function:*

$$\max_{p_\theta} \min_{\boldsymbol{\lambda} \in \mathbb{R}_+^N} \mathcal{J}^{(\mathcal{L})}(p_\theta, \boldsymbol{\lambda}) = \mathcal{J}^{(\mathcal{L})}(p^*, \boldsymbol{\lambda}^*) = \min_{\boldsymbol{\lambda} \in \mathbb{R}_+^N} \max_{p_\theta} \mathcal{J}^{(\mathcal{L})}(p_\theta, \boldsymbol{\lambda}). \tag{13}$$

Specific proofs can be found in previous works (Liu et al., 2024; Huang et al., 2024).

### A.3.2 CONTINUOUS REWARD PREFERENCE OPTIMIZATION FOR SCORE-BASED DIFFUSION MODELS

In this section, we derive the training objective for the score-based diffusion model using a continuous reward from the objective form of RLHF. Following the approach of previous work (Wallace et al., 2024), we first define a reparameterization of reward. Subsequently, based on this reward, we can obtain the reparameterization of $f_\theta$ as described in (Chen et al., 2024a).

We simplify the notation in this section using $q$, $p_\theta$, and $p_{\text{ref}}$ to represent $q^{x,r,a}$, $p_\theta^{x,r,a}$, and $p_{\text{ref}}^{x,r,a}$, respectively.

We define $r_m(\mathbf{T}^{(0:1)}, \mathcal{P}_{\text{FC}})$ and $c_n(\mathbf{T}^{(0:1)}, \mathcal{P}_{\text{FC}})$ as the reward function and the constraint function on the whole trajectory of diffusion process, such that we define $\mathcal{R}_m(\mathbf{T}^{(0)}, \mathcal{P}_{\text{FC}})$ and $\mathcal{C}_n(\mathbf{T}^{(0)}, \mathcal{P}_{\text{FC}})$ as follows:

$$\begin{aligned}
\mathcal{R}_m(\mathbf{T}^{(0)}, \mathcal{P}_{\text{FC}}) &= \mathbb{E}_{\mathbf{T}^{(\Delta t:1)} \sim p_\theta(\mathbf{T}^{(\Delta t:1)} | \mathbf{T}^{(0)}, \mathcal{P}_{\text{FC}})}[r_m(\mathbf{T}^{(0:1)}, \mathcal{P}_{\text{FC}})], \\
\mathcal{C}_n(\mathbf{T}^{(0)}, \mathcal{P}_{\text{FC}}) &= \mathbb{E}_{\mathbf{T}^{(\Delta t:1)} \sim p_\theta(\mathbf{T}^{(\Delta t:1)} | \mathbf{T}^{(0)}, \mathcal{P}_{\text{FC}})}[c_n(\mathbf{T}^{(0:1)}, \mathcal{P}_{\text{FC}})],
\end{aligned} \tag{14}$$

where $\mathbf{T}^{(0:1)} = (\mathbf{T}^{(0)}, \mathbf{T}^{(\Delta t)}, ..., \mathbf{T}^{(1)})$ means the inference trajectory of diffusion process. Then, we only consider per $\mathcal{P}_{\text{FC}}$ for simplification and the objective of AbNovo can be written as follows:

$$\begin{aligned}
\max_\theta \mathbb{E}_{\mathbf{T}^{(0)} \sim p_\theta(\mathbf{T}^{(0)} | \mathcal{P}_{\text{FC}})} \Bigg[ &\sum_m \omega_m \mathcal{R}_m(\mathbf{T}^{(0)}, \mathcal{P}_{\text{FC}}) - \sum_n \lambda_n \mathcal{C}_n(\mathbf{T}^{(0)}, \mathcal{P}_{\text{FC}}) + \sum_n \lambda_n C_n \Bigg] \\
&- \beta \mathbb{D}_{\text{KL}}(p_\theta(\mathbf{T}^{(0)} | \mathcal{P}_{\text{FC}}) || p_{\text{ref}}(\mathbf{T}^{(0)} | \mathcal{P}_{\text{FC}})).
\end{aligned} \tag{15}$$

From Equation 7, we can obtain $\hat{\mathcal{R}}(\mathbf{T}^{(0)}, \mathcal{P}_{\text{FC}})$ as

$$\hat{\mathcal{R}}(\mathbf{T}^{(0)}, \mathcal{P}_{\text{FC}}) = \mathbb{E}_{\mathbf{T}^{(\Delta t:1)} \sim p_\theta(\mathbf{T}^{(\Delta t:1)} | \mathbf{T}^{(0)}, \mathcal{P}_{\text{FC}})}[\hat{r}(\mathbf{T}^{(0:1)}, \mathcal{P}_{\text{FC}})], \tag{16}$$

where $\hat{r}(\mathbf{T}^{(0:1)}, \mathcal{P}_{\text{FC}})$ is defined as

$$\hat{r}(\mathbf{T}^{(0:1)}, \mathcal{P}_{\text{FC}}) = \sum_m \omega_m r_m(\mathbf{T}^{(0:1)}, \mathcal{P}_{\text{FC}}) - \sum_n \lambda_n c_n(\mathbf{T}^{(0:1)}, \mathcal{P}_{\text{FC}}). \tag{17}$$

Then, the objective of 15 can be transformed to the following form:

$$
\min_\theta - \mathbb{E}_{\mathbf{T}^{(0)}\sim p_\theta(\mathbf{T}^{(0)}|\mathcal{P}_{\mathrm{FC}}))} \Big[ \hat{\mathcal{R}}(\mathbf{T}^{(0)}, \mathcal{P}_{\mathrm{FC}}) + \sum_n \lambda_n C_n \Big]/\beta
$$
$$
+ \mathbb{D}_{\mathrm{KL}}(p_\theta(\mathbf{T}^{(0)}|\mathcal{P}_{\mathrm{FC}})||p_{\mathrm{ref}}(\mathbf{T}^{(0)}|\mathcal{P}_{\mathrm{FC}}))
$$
$$
\leq \min_\theta - \mathbb{E}_{\mathbf{T}^{(0)}\sim p_\theta(\mathbf{T}^{(0)}|\mathcal{P}_{\mathrm{FC}}))} \Big[ \hat{\mathcal{R}}(\mathbf{T}^{(0)}, \mathcal{P}_{\mathrm{FC}}) + \sum_n \lambda_n C_n \Big]/\beta
$$
$$
+ \mathbb{D}_{\mathrm{KL}}(p_\theta(\mathbf{T}^{(0:1)}|\mathcal{P}_{\mathrm{FC}})||p_{\mathrm{ref}}(\mathbf{T}^{(0:1)}|\mathcal{P}_{\mathrm{FC}}))
$$
$$
= \min_\theta - \mathbb{E}_{\mathbf{T}^{(0:1)}\sim p_\theta(\mathbf{T}^{(0:1)}|\mathcal{P}_{\mathrm{FC}}))} \Big[ \hat{r}(\mathbf{T}^{(0:1)}, \mathcal{P}_{\mathrm{FC}}) \Big]/\beta
$$
$$
+ \mathbb{D}_{\mathrm{KL}}(p_\theta(\mathbf{T}^{(0:1)}|\mathcal{P}_{\mathrm{FC}})||p_{\mathrm{ref}}(\mathbf{T}^{(0:1)}|\mathcal{P}_{\mathrm{FC}})) - \frac{1}{\beta}\sum_n \lambda_n C_n
$$
$$
= \min_\theta \mathbb{E}_{\mathbf{T}^{(0:1)}\sim p_\theta(\mathbf{T}^{(0:1)}|\mathcal{P}_{\mathrm{FC}}))} \Bigg[ \log \frac{p_\theta(\mathbf{T}^{(0:1)}|\mathcal{P}_{\mathrm{FC}})}{p_{\mathrm{ref}}(\mathbf{T}^{(0:1)}|\mathcal{P}_{\mathrm{FC}})\exp(\hat{r}(\mathbf{T}^{(0:1)}, \mathcal{P}_{\mathrm{FC}})/\beta)/Z(\mathcal{P}_{\mathrm{FC}})}
$$
$$
- \log Z(\mathcal{P}_{\mathrm{FC}}) \Bigg] - \frac{1}{\beta}\sum_n \lambda_n C_n,
$$

$$(18)$$

where $Z(\mathcal{P}_{\mathrm{FC}}) = \mathbb{E}_{\mathbf{T}^{(0:1)}\sim p_{\mathrm{ref}}(\mathbf{T}^{(0:1)}|\mathcal{P}_{\mathrm{FC}})} \exp(\hat{r}(\mathbf{T}^{(0:1)}, \mathcal{P}_{\mathrm{FC}})/\beta)$ and $C_n$ is independent of $\theta$. The optimal $p_\theta^*(\mathbf{T}^{(0:1)}|\mathcal{P}_{\mathrm{FC}})$ of Equation 18 has a unique closed-form solution:

$$
p_\theta^*(\mathbf{T}^{(0:1)}|\mathcal{P}_{\mathrm{FC}}) = p_{\mathrm{ref}}(\mathbf{T}^{(0:1)}|\mathcal{P}_{\mathrm{FC}}) \exp(\hat{r}(\mathbf{T}^{(0:1)}, \mathcal{P}_{\mathrm{FC}})/\beta)/Z(\mathcal{P}_{\mathrm{FC}}). \tag{19}
$$

Therefore, we have the reparameterization of $\hat{r}(\mathbf{T}^{(0:1)}, \mathcal{P}_{\mathrm{FC}})$ as follows:

$$
\hat{r}(\mathbf{T}^{(0:1)}, \mathcal{P}_{\mathrm{FC}}) = \beta \log \frac{p_\theta^*(\mathbf{T}^{(0:1)}|\mathcal{P}_{\mathrm{FC}})}{p_{\mathrm{ref}}(\mathbf{T}^{(0:1)}|\mathcal{P}_{\mathrm{FC}})} + \beta \log Z(\mathcal{P}_{\mathrm{FC}}). \tag{20}
$$

Plug this into the definition of $\hat{\mathcal{R}}(\mathbf{T}^{(0)}, \mathcal{P}_{\mathrm{FC}})$ of Equation 16, hence we have:

$$
\hat{\mathcal{R}}(\mathbf{T}^{(0)}, \mathcal{P}_{\mathrm{FC}}) = \beta \mathbb{E}_{\mathbf{T}^{(\Delta t:1)}\sim p_\theta(\mathbf{T}^{(\Delta t:1)}|\mathbf{T}^{(0)}, \mathcal{P}_{\mathrm{FC}})} \Bigg[ \log \frac{p_\theta^*(\mathbf{T}^{(0:1)}|\mathcal{P}_{\mathrm{FC}})}{p_{\mathrm{ref}}(\mathbf{T}^{(0:1)}|\mathcal{P}_{\mathrm{FC}})} \Bigg] + \beta \log Z(\mathcal{P}_{\mathrm{FC}}). \tag{21}
$$

Then by Equation (15) in (Chen et al., 2024a), the $f_\theta$ can be defined as follows:

$$
f_\theta(\mathbf{T}^{(0)}, \mathcal{P}_{\mathrm{FC}}) = \mathbb{E}_{\mathbf{T}^{(\Delta t:1)}\sim p_\theta(\mathbf{T}^{(\Delta t:1)}|\mathbf{T}^{(0)}, \mathcal{P}_{\mathrm{FC}})} \Bigg[ \log \frac{p_\theta(\mathbf{T}^{(0:1)}|\mathcal{P}_{\mathrm{FC}})}{p_{\mathrm{ref}}(\mathbf{T}^{(0:1)}|\mathcal{P}_{\mathrm{FC}})} \Bigg]. \tag{22}
$$

Here, considering that the computation of $Z(\mathcal{P}_{\mathrm{FC}})$ is intractable, we approximate it by $\mathbb{E}_{\mathbf{T}^{(0)}\sim p_{\mathrm{ref}}(\mathbf{T}^{(0)}|\mathcal{P}_{\mathrm{FC}})} \exp(\hat{\mathcal{R}}(\mathbf{T}^0, \mathcal{P}_{\mathrm{FC}})/\beta)$ to simplify the computation.

Substituting this reparameterization of $f_\theta$ into Equation (16) in Chen et al. (2024a), we obtain the objective of diffusion-based NCA:

$$
\mathcal{L}_{\mathrm{NCA}}^{\mathrm{diff}}(\theta) = - \sum_{i=1}^K \Bigg[ \frac{\exp(\hat{\mathcal{R}}_i/\beta)}{\sum\limits_{j=1}^K \exp(\hat{\mathcal{R}}_j/\beta)} \log \sigma\Big( f_\theta(\mathbf{T}_i^{(0)}, \mathcal{P}_{\mathrm{FC}}) \Big) + \frac{1}{K} \log \sigma\Big( -f_\theta(\mathbf{T}_i^{(0)}, \mathcal{P}_{\mathrm{FC}}) \Big) \Bigg]
$$
$$
= - \sum_{i=1}^K \Bigg[ \frac{\exp(\hat{\mathcal{R}}_i/\beta)}{\sum\limits_{j=1}^K \exp(\hat{\mathcal{R}}_j/\beta)} \log \sigma\Bigg( \mathbb{E}_{\mathbf{T}_i^{(\Delta t:1)}\sim p_\theta(\mathbf{T}_i^{(\Delta t:1)}|\mathbf{T}_i^{(0)}, \mathcal{P}_{\mathrm{FC}})} \Bigg[ \log \frac{p_\theta(\mathbf{T}_i^{(0:1)}|\mathcal{P}_{\mathrm{FC}})}{p_{\mathrm{ref}}(\mathbf{T}_i^{(0:1)}|\mathcal{P}_{\mathrm{FC}})} \Bigg] \Bigg)
$$
$$
+ \frac{1}{K} \log \sigma\Bigg( -\mathbb{E}_{\mathbf{T}_i^{(\Delta t:1)}\sim p_\theta(\mathbf{T}_i^{(\Delta t:1)}|\mathbf{T}_i^{(0)}, \mathcal{P}_{\mathrm{FC}})} \Bigg[ \log \frac{p_\theta(\mathbf{T}_i^{(0:1)}|\mathcal{P}_{\mathrm{FC}})}{p_{\mathrm{ref}}(\mathbf{T}_i^{(0:1)}|\mathcal{P}_{\mathrm{FC}})} \Bigg] \Bigg) \Bigg].
$$

$$(23)$$

Since sampling from $p_\theta(\mathbf{T}_i^{(\Delta t:1)}|\mathbf{T}_i^{(0)}, \mathcal{P}_{\text{FC}})$ is intractable, we utilize $q(\mathbf{T}_i^{(\Delta t:1)}|\mathbf{T}_i^{(0)}, \mathcal{P}_{\text{FC}})$ for approximathon (Wallace et al., 2024).

$$
\begin{aligned}
\mathcal{L}_{\text{NCA}}^{\text{diff}}(\theta) = &-\sum_{i=1}^{K} \left[ \frac{\exp(\hat{\mathcal{R}}_i/\beta)}{\sum_{j=1}^{K} \exp(\hat{\mathcal{R}}_j/\beta)} \log \sigma \left( \mathbb{E}_{\mathbf{T}_i^{(\Delta t:1)} \sim q(\mathbf{T}_i^{(\Delta t:1)}|\mathbf{T}_i^{(0)}, \mathcal{P}_{\text{FC}})} \left[ \log \frac{p_\theta(\mathbf{T}_i^{(0:1)}|\mathcal{P}_{\text{FC}})}{p_{\text{ref}}(\mathbf{T}_i^{(0:1)}|\mathcal{P}_{\text{FC}})} \right] \right) \right. \\
&\left. + \frac{1}{K} \log \sigma \left( -\mathbb{E}_{\mathbf{T}_i^{(\Delta t:1)} \sim q(\mathbf{T}_i^{(\Delta t:1)}|\mathbf{T}_i^{(0)}, \mathcal{P}_{\text{FC}})} \left[ \log \frac{p_\theta(\mathbf{T}_i^{(0:1)}|\mathcal{P}_{\text{FC}})}{p_{\text{ref}}(\mathbf{T}_i^{(0:1)}|\mathcal{P}_{\text{FC}})} \right] \right) \right] \\
= &-\sum_{i=1}^{K} \left[ \frac{\exp(\hat{\mathcal{R}}_i/\beta)}{\sum_{j=1}^{K} \exp(\hat{\mathcal{R}}_j/\beta)} \log \sigma \left( \mathbb{E}_{\mathbf{T}_i^{(\Delta t:1)} \sim q(\mathbf{T}_i^{(\Delta t:1)}|\mathbf{T}_i^{(0)}, \mathcal{P}_{\text{FC}})} \left[ \sum_{\substack{t \in \{\Delta t, \\ \dots, 1\}}} \log \frac{p_\theta(\mathbf{T}_i^{(t-\Delta t)}|\mathbf{T}_i^{(t)}, \mathcal{P}_{\text{FC}})}{p_{\text{ref}}(\mathbf{T}_i^{(t-\Delta t)}|\mathbf{T}_i^{(t)}, \mathcal{P}_{\text{FC}})} \right] \right) \right. \\
&\left. + \frac{1}{K} \log \sigma \left( -\mathbb{E}_{\mathbf{T}_i^{(\Delta t:1)} \sim q(\mathbf{T}_i^{(\Delta t:1)}|\mathbf{T}_i^{(0)}, \mathcal{P}_{\text{FC}})} \left[ \sum_{\substack{t \in \{\Delta t, \\ \dots, 1\}}} \log \frac{p_\theta(\mathbf{T}_i^{(t-\Delta t)}|\mathbf{T}_i^{(t)}, \mathcal{P}_{\text{FC}})}{p_{\text{ref}}(\mathbf{T}_i^{(t-\Delta t)}|\mathbf{T}_i^{(t)}, \mathcal{P}_{\text{FC}})} \right] \right) \right] \\
= &-\sum_{i=1}^{K} \left[ \frac{\exp(\hat{\mathcal{R}}_i/\beta)}{\sum_{j=1}^{K} \exp(\hat{\mathcal{R}}_j/\beta)} \log \sigma \left( \mathbb{E}_{\mathbf{T}_i^{(\Delta t:1)} \sim q(\mathbf{T}_i^{(\Delta t:1)}|\mathbf{T}_i^{(0)}, \mathcal{P}_{\text{FC}})} \frac{1}{\Delta t} \mathbb{E}_t \left[ \log \frac{p_\theta(\mathbf{T}_i^{(t-\Delta t)}|\mathbf{T}_i^{(t)}, \mathcal{P}_{\text{FC}})}{p_{\text{ref}}(\mathbf{T}_i^{(t-\Delta t)}|\mathbf{T}_i^{(t)}, \mathcal{P}_{\text{FC}})} \right] \right) \right. \\
&\left. + \frac{1}{K} \log \sigma \left( -\mathbb{E}_{\mathbf{T}_i^{(\Delta t:1)} \sim q(\mathbf{T}_i^{(\Delta t:1)}|\mathbf{T}_i^{(0)}, \mathcal{P}_{\text{FC}})} \frac{1}{\Delta t} \mathbb{E}_t \left[ \log \frac{p_\theta(\mathbf{T}_i^{(t-\Delta t)}|\mathbf{T}_i^{(t)}, \mathcal{P}_{\text{FC}})}{p_{\text{ref}}(\mathbf{T}_i^{(t-\Delta t)}|\mathbf{T}_i^{(t)}, \mathcal{P}_{\text{FC}})} \right] \right) \right] \\
= &-\sum_{i=1}^{K} \left[ \frac{\exp(\hat{\mathcal{R}}_i/\beta)}{\sum_{j=1}^{K} \exp(\hat{\mathcal{R}}_j/\beta)} \log \sigma \left( \frac{1}{\Delta t} \mathbb{E}_t \mathbb{E}_{\mathbf{T}_i^{(t)} \sim q(\mathbf{T}_i^{(t)}|\mathbf{T}_i^{(0)}, \mathcal{P}_{\text{FC}})} \right. \right. \\
&\left. \mathbb{E}_{\mathbf{T}_i^{(t-\Delta t)} \sim q(\mathbf{T}_i^{(t-\Delta t)}|\mathbf{T}_i^{(t)}, \mathbf{T}_i^{(0)}, \mathcal{P}_{\text{FC}})} \left[ \log \frac{p_\theta(\mathbf{T}_i^{(t-\Delta t)}|\mathbf{T}_i^{(t)}, \mathcal{P}_{\text{FC}})}{p_{\text{ref}}(\mathbf{T}_i^{(t-\Delta t)}|\mathbf{T}_i^{(t)}, \mathcal{P}_{\text{FC}})} \right] \right) \\
&\left. + \frac{1}{K} \log \sigma \left( -\frac{1}{\Delta t} \mathbb{E}_t \mathbb{E}_{\mathbf{T}_i^{(t)} \sim q(\mathbf{T}_i^{(t)}|\mathbf{T}_i^{(0)}, \mathcal{P}_{\text{FC}})} \right. \right. \\
&\left. \left. \mathbb{E}_{\mathbf{T}_i^{(t-\Delta t)} \sim q(\mathbf{T}_i^{(t-\Delta t)}|\mathbf{T}_i^{(t)}, \mathbf{T}_i^{(0)}, \mathcal{P}_{\text{FC}})} \left[ \log \frac{p_\theta(\mathbf{T}_i^{(t-\Delta t)}|\mathbf{T}_i^{(t)}, \mathcal{P}_{\text{FC}})}{p_{\text{ref}}(\mathbf{T}_i^{(t-\Delta t)}|\mathbf{T}_i^{(t)}, \mathcal{P}_{\text{FC}})} \right] \right) \right].
\end{aligned}
\tag{24}
$$

By using Jensen's inequality, we have:

$$
\begin{aligned}
\mathcal{L}_{\text{NCA}}^{\text{diff}}(\theta) \leq -\sum_{i=1}^{K} & \left[ \frac{\exp\left(\hat{\mathcal{R}}_i/\beta\right)}{\sum_{j=1}^{K} \exp\left(\hat{\mathcal{R}}_j/\beta\right)} \mathbb{E}_t \mathbb{E}_{\mathbf{T}_i^{(t)} \sim q(\mathbf{T}_i^{(t)}|\mathbf{T}_i^{(0)},\mathcal{P}_{\text{FC}})} \log \sigma \left( \vphantom{\frac{1}{\Delta t}} \right.\right. \\
& \frac{1}{\Delta t} \mathbb{E}_{\mathbf{T}_i^{(t-\Delta t)} \sim q(\mathbf{T}_i^{(t-\Delta t)}|\mathbf{T}_i^{(t)},\mathbf{T}_i^{(0)},\mathcal{P}_{\text{FC}})} \left[ \log \frac{p_\theta(\mathbf{T}_i^{(t-\Delta t)}|\mathbf{T}_i^{(t)},\mathcal{P}_{\text{FC}})}{p_{\text{ref}}(\mathbf{T}_i^{(t-\Delta t)}|\mathbf{T}_i^{(t)},\mathcal{P}_{\text{FC}})} \right] \right) \\
& + \frac{1}{K} \mathbb{E}_t \mathbb{E}_{\mathbf{T}_i^{(t)} \sim q(\mathbf{T}_i^{(t)}|\mathbf{T}_i^{(0)},\mathcal{P}_{\text{FC}})} \log \sigma \left( \vphantom{\frac{1}{\Delta t}} \right. \\
& \left. \left. - \frac{1}{\Delta t} \mathbb{E}_{\mathbf{T}_i^{(t-\Delta t)} \sim q(\mathbf{T}_i^{(t-\Delta t)}|\mathbf{T}_i^{(t)},\mathbf{T}_i^{(0)},\mathcal{P}_{\text{FC}})} \left[ \log \frac{p_\theta(\mathbf{T}_i^{(t-\Delta t)}|\mathbf{T}_i^{(t)},\mathcal{P}_{\text{FC}})}{p_{\text{ref}}(\mathbf{T}_i^{(t-\Delta t)}|\mathbf{T}_i^{(t)},\mathcal{P}_{\text{FC}})} \right] \right) \right] \\
= -\mathbb{E}_t \mathbb{E}_{\{\mathbf{T}_i^{(t)} \sim q(\mathbf{T}_i^{(t)}|\mathbf{T}_i^{(0)},\mathcal{P}_{\text{FC}})\}_{1:K}} & \sum_{i=1}^{K} \left[ \frac{\exp\left(\hat{\mathcal{R}}_i/\beta\right)}{\sum_{j=1}^{K} \exp\left(\hat{\mathcal{R}}_j/\beta\right)} \log \sigma \left( \vphantom{\frac{1}{\Delta t}} \right.\right. \\
& \frac{1}{\Delta t} \mathbb{E}_{\mathbf{T}_i^{(t-\Delta t)} \sim q(\mathbf{T}_i^{(t-\Delta t)}|\mathbf{T}_i^{(t)},\mathbf{T}_i^{(0)},\mathcal{P}_{\text{FC}})} \left[ \log \frac{p_\theta(\mathbf{T}_i^{(t-\Delta t)}|\mathbf{T}_i^{(t)},\mathcal{P}_{\text{FC}})}{p_{\text{ref}}(\mathbf{T}_i^{(t-\Delta t)}|\mathbf{T}_i^{(t)},\mathcal{P}_{\text{FC}})} \right] \right) \\
& + \frac{1}{K} \log \sigma \left( -\frac{1}{\Delta t} \mathbb{E}_{\mathbf{T}_i^{(t-\Delta t)} \sim q(\mathbf{T}_i^{(t-\Delta t)}|\mathbf{T}_i^{(t)},\mathbf{T}_i^{(0)},\mathcal{P}_{\text{FC}})} \left[ \log \frac{p_\theta(\mathbf{T}_i^{(t-\Delta t)}|\mathbf{T}_i^{(t)},\mathcal{P}_{\text{FC}})}{p_{\text{ref}}(\mathbf{T}_i^{(t-\Delta t)}|\mathbf{T}_i^{(t)},\mathcal{P}_{\text{FC}})} \right] \right) \right].
\end{aligned}
\tag{25}
$$

With some algebra, the above loss simplifies to Equation 9.

### A.3.3 OFFLINE DUAL GRADIENT ESTIMATES

In this section, we introduce how to use *offline* data to estimate the gradient of the dual function.

From Equation 11, the gradient of the dual function $\mathcal{G}(\boldsymbol{\lambda})$ can be calculated as follows:

$$
\begin{aligned}
\frac{\mathrm{d}\mathcal{G}(\boldsymbol{\lambda})}{\mathrm{d}\boldsymbol{\lambda}} &= \frac{\mathrm{d}\mathcal{J}^{(L)}(p_{\boldsymbol{\lambda}}^*,\boldsymbol{\lambda})}{\mathrm{d}p_{\boldsymbol{\lambda}}^*} \frac{\mathrm{d}p_{\boldsymbol{\lambda}}^*}{\mathrm{d}\boldsymbol{\lambda}} + \frac{\mathcal{J}^{(L)}(p_{\boldsymbol{\lambda}}^*,\boldsymbol{\lambda})}{\mathrm{d}\boldsymbol{\lambda}} \\
&= \frac{\mathrm{d}\mathcal{J}^{(L)}(p_{\boldsymbol{\lambda}}^*,\boldsymbol{\lambda})}{\mathrm{d}\boldsymbol{\lambda}} \\
&= \frac{\mathrm{d}\left(\mathcal{J}^{(\mathcal{R})}(p_{\boldsymbol{\lambda}}^*) - \mathcal{J}^{(\mathcal{C})}(p_{\boldsymbol{\lambda}}^*,\boldsymbol{\lambda})\right)}{\mathrm{d}\boldsymbol{\lambda}} \\
&= -\frac{\mathrm{d}\mathcal{J}^{(\mathcal{C})}(p_{\boldsymbol{\lambda}}^*,\boldsymbol{\lambda})}{\mathrm{d}\boldsymbol{\lambda}} \\
&= \mathbb{E}_{\mathcal{P}_{\text{FC}} \in \mathcal{D},\mathbf{T}^{(0)} \sim p_{\boldsymbol{\lambda}}^*(\mathbf{T}^{(0)}|\mathcal{P}_{\text{FC}})} \left[ \mathbf{C} - \boldsymbol{\mathcal{C}}(\mathbf{T}^{(0)},\mathcal{P}_{\text{FC}}) \right],
\end{aligned}
\tag{26}
$$

where $\boldsymbol{\lambda} = [\lambda_1, \lambda_2, ..., \lambda_n]$. After updating policy, the optimal solution of Equation 9 under $\boldsymbol{\lambda}$ is

$$
p_{\boldsymbol{\lambda}}^* \propto p_{\text{ref}} \exp(\hat{\mathcal{R}}/\beta).
\tag{27}
$$

Thus, Equation 26 could be written as

$$
\frac{\mathrm{d}\mathcal{G}(\boldsymbol{\lambda})}{\mathrm{d}\boldsymbol{\lambda}} = \mathbb{E}_{\mathcal{P}_{\text{FC}} \in \mathcal{D}_g} \left[ \frac{\mathbb{E}_{\mathbf{T}^{(0)} \sim p_{\text{ref}}(\mathbf{T}^{(0)}|\mathcal{P}_{\text{FC}})}\left( \exp(\hat{\mathcal{R}}/\beta)(\mathbf{C} - \boldsymbol{\mathcal{C}}(\mathbf{T}^{(0)},\mathcal{P}_{\text{FC}})) \right)}{\mathbb{E}_{\mathbf{T}^{(0)} \sim p_{\text{ref}}(\mathbf{T}^{(0)}|\mathcal{P}_{\text{FC}})} \exp(\hat{\mathcal{R}}/\beta)} \right],
\tag{28}
$$

Then, the gradient of $\mathcal{G}(\boldsymbol{\lambda})$ could be estimated by *offline* data, since Equation 28 only correlated by $p_{\text{ref}}$. Therefore, we use a large *offline* dataset $\mathcal{D}_g = \{\{\mathcal{R}_m(\mathbf{T}_{j,k}^{(0)},\mathcal{P}_{\text{FC},k}),\mathcal{C}_n(\mathbf{T}_{j,k}^{(0)},\mathcal{P}_{\text{FC},k})\}_{j=1}^{J}\}_{k=1}^{K}$

(for $K$ antigens, $J$ antibodies were sampled for each antigen) to estimate the gradient. The specific form can be written as follows:

$$\frac{\mathrm{d}\mathcal{G}(\boldsymbol{\lambda})}{\mathrm{d}\boldsymbol{\lambda}} = -\frac{1}{|\mathcal{D}_g|}\sum_{k=1}^{K}\sum_{j=1}^{J}\mathrm{softmax}\Big(\big\{\hat{\mathcal{R}}_j/\beta\big\}_{j=1}^{J}\Big)_j \boldsymbol{\mathcal{C}}(\mathbf{T}_{j,k}^{(0)}, \mathcal{P}_{\mathrm{FC},k}) + \boldsymbol{\mathcal{C}}_{\mathrm{avg}} + \mathbf{C}, \qquad (29)$$

where $\hat{\mathcal{R}}$ is defined as Equation 7, $\boldsymbol{\mathcal{C}} = [\mathcal{C}_1, ..., \mathcal{C}_n]$ is the constraint functions, $\mathbf{C} = [C_1, ..., C_n]$ is the thresholds for different constraints, and $\boldsymbol{\mathcal{C}}_{\mathrm{avg}}$ is the global normalization term.

Here, we use $\{.\}_i$ to denote the $i$-th element of the vector. If the current $\lambda_j$ is higher, while the model already satisfies the $j$-th constraint. Thus, a higher $\lambda_j$ will result in a greater weight for samples with high $\mathcal{G}_j$, and $\{\frac{\mathrm{d}\mathcal{G}(\boldsymbol{\lambda})}{\mathrm{d}\boldsymbol{\lambda}}\}_j$ is larger. Eventually, the gradient of the $\lambda_j$ will increase. Conversely, if the current $\lambda_j$ is low, while the model appears to violate the $j$-th constraint. Such a lower $\lambda_j$ will result in a smaller weight for samples with low $\mathcal{G}_j$, and $\{\frac{\mathrm{d}\mathcal{G}(\boldsymbol{\lambda})}{\mathrm{d}\boldsymbol{\lambda}}\}_j$ will be smaller, and after being normalized, the gradient of the $\lambda_j$ will be negative at this step; the next step will increase the $\lambda_j$.

## A.4 ABNOVO BASE MODEL

This section introduces AbNovo's training strategies, training losses, and inference processes.

### A.4.1 PRE-TRAINED LANGUAGE MODEL

We targeted masking $10\%$ of the protein sequences using the BERT (Lin et al., 2023) model. Specifically, $85\%$ of the masked positions were replaced with the [mask] token, $10\%$ were substituted with random amino acids, and $5\%$ remained unchanged. Additionally, for protein sequences longer than 200 residues, we randomly masked consecutive amino acid segments of length between 5 and 13. Concurrently, we predicted the distance matrix of amino acids using the distogram loss $\mathcal{L}$distogram for input sequences containing masked regions. We also employed a contact prediction loss function $\mathcal{L}$contact, considering amino acids with distances greater than 8 Å as long-distance contacts. The ESM2-3B model (Lin et al., 2023) was used as our model architecture, with its pre-trained weights serving as initialization.

### A.4.2 DIFFUSION PROCESS DETAILS

Following the setup from previous work (Campbell et al., 2024), our noise schedules are independently tailored for each of the three diffusion processes (Table 11).

For diffusion computation and parameter settings, we follow previous works (Yim et al., 2023; Zhu et al., 2024).

Table 11: Noise schedule settings.

| Translation | $\beta(t) = \beta_{\min} + t(\beta_{\max} - \beta_{\min})$ | $\beta_{\min} = 0.1; \beta_{\max} = 20$ |
|---|---|---|
| Rotation | $\sigma(t) = \log(t\exp(\sigma_{\max}) + (1-t)\exp(\sigma_{\min}))$ | $\sigma_{\min} = 0.01; \sigma_{\max} = 2.25$ |
| Sequence | $\alpha(t) = \frac{1}{3(1-t)}$ | - |

### A.4.3 ABNOVO MODEL ARCHITECTURE

**Networks.** AbNovo adopts a network structure similar to ESMFold (Lin et al., 2023), comprising our pre-trained structure-aware language model, a 4-layer `Main Trunk`, IPA, a structure encoder and a sequence decoder. Our sequence encoder is a structure-aware language model with 33 `Transformer` layers, while the structure encoder is a `Linear` layer. The sequence decoder maps the single representation obtained from the `Main Trunk` to amino acid species, consisting of three layers of `MLPs`. The specific network dimensions are provided in Table 14.

The model architecture of AbNovo is depicted in Figure 4. Consistent with prior studies (Jumper et al., 2021; Lin et al., 2023; Zhu et al., 2024; Abramson et al., 2024), we find that employing the

recycling technique can further enhance performance. The recycling features include the predicted antibody sequence and the antibody distance matrix.

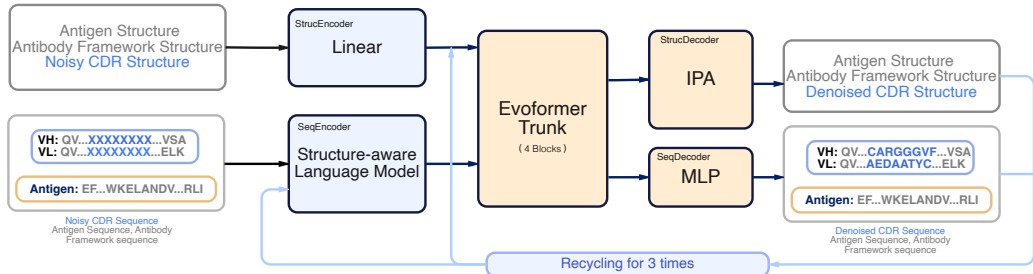

Figure 4: Encoder and score network of AbNovo.

**Input Features.** The specific details of the input features are shown in Table 12.

Table 12: Input features of AbNovo.

| Modules | Description of Input Features |
|---|---|
| Sequence Encoder | 1. Sequence of antibody framework, antigen, and noisy CDR |
| Structure Encoder | 1. Distance matrix of antibody framework, antigen, and noisy CDR
2. Backbone dihedral angles, including $\phi, \psi$ and $\omega$ in sine and cosine form of antibody framework, antigen and noisy CDR |
| Others | 1. Chain ID of antibody and antigen
2. Time embedding of Sequence and Structure
3. Relative position embedding |

Table 13: Network hyper-parameter of AbNovo.

| Description | Value |
|---|---|
| dimension of single representation in `Evoformer Trunk` | 512 |
| dimension of pair representation in `Evoformer Trunk` | 128 |
| dimension of single representation in `IPA` | 384 |
| dimension of sequence decoder | 512 |

### A.4.4 DATASETS

**Datasets for Structure-aware Language Model.** In the training process, we used all the proteins from the Protein Data Bank (PDB) before 2020-01-01 for training, except that we filtered some antibody data. For antibody structure data, we filtered for the CDR H3 region in the training set with more than $40\%$ sequence similarity with the test set. In the training process, we followed the data processing method of AlphaFold2 (Jumper et al., 2021), and we used MMseqs for $40\%$ sequence similarity clustering.

For proteins in AlphaFold DataBase (AFDB), we masked all amino acids with a pLDDT of less than 50 and removed proteins with an overall pLDDT of less than 70. During the training process, we maintained a crop size 512 and kept the sampling probability of the PDB training data and the training data from the AFDB at 1:4. We used the parameters of the ESM2-3B (Lin et al., 2023) model as initialization.

**Datasets for Training base model.** We follow the list of provided training sets from previous work (Luo et al., 2022) in `https://github.com/luost26/diffab/blob/main/data/sabdab_summary_all.tsv`.

**Datasets for Preference Optimization.** Following previous works (Zhou et al., 2024), we construct preference-optimized datasets for antigen and antibody complexes. We generate 512 antibod-

ies for each antigen in each preference optimization process, and then the metrics are calculated for these antibodies.

When updating $\lambda$, we randomly selected 10000 (for 50 antigens and 200 of each antibody) antigen-antibody complexes to estimate the gradient.

### A.4.5 TRAINING LOSSES

**Training Losses for Structure-aware Language Model.** We use Mask Language Model Loss (Lin et al., 2023), distogram loss (Jumper et al., 2021), and contact prediction loss to train our base model. The specific loss function is as follows:

$$\mathcal{L}_{\text{pretrain}} = 0.5\mathcal{L}_{\text{MLM}} + 1.0\mathcal{L}_{\text{distogram}} + 1.0\mathcal{L}_{\text{contact}},$$

$$\mathcal{L}_{\text{distogram}} = -\frac{1}{L^2} \sum_{i,j} \sum_{b=1}^{b=32} y_{ij}^b \log p_{ij}^b, \tag{30}$$

$$\mathcal{L}_{\text{contact}} = -\frac{1}{L^2} \sum_{i,j} \sum_{b \in \Omega} \mathbb{I}(d_{ij} < 8\text{Å}) \log p_{ij}^b.$$

Distogram loss ($\mathcal{L}_{\text{distogram}}$) is the prediction of distances between amino acid pairs. Specifically, we divide the distance from 2Å to 22Å into 32 bins and predict in which bin the distance to the amino acid pair lies, where $p_{ij}^b$ is the probability of each bin. Contact loss ($\mathcal{L}_{\text{contact}}$) is the prediction of whether amino acid pairs are in contact or not (whether the distance is less than 8Å or not), where $\Omega \in \{\text{contact}, \text{not contact}\}$.

**Training Losses for Base Model.** We use four loss for training AbNovo (base) $\mathcal{L}^{(x)}, \mathcal{L}^{(r)}, \mathcal{L}^{(a)}, \mathcal{L}_{\text{violation}}$ and $\mathcal{L}_{\text{aux}}$. Following previous works (Yim et al., 2023; Campbell et al., 2024), the denoising score matching (DSM) of different modal can be written as follows:

$$\mathcal{L}^{(r)} = \frac{1}{L} \sum_{i=1}^{L} \alpha_t ||\nabla_r \log p_{t|0}(r_i^{(t)}|\hat{r}_i^{(0)}) - \nabla_r p_{t|0}(r_i^{(t)}|r_i^{(0)})||_2^2,$$

$$\mathcal{L}^{(x)} = \frac{1}{L} \sum_{i=1}^{L} ||\hat{x}_i^{(0)} - x_i^{(0)}||_2^2, \tag{31}$$

$$\mathcal{L}^{(a)} = \frac{1}{L} \sum_{i=1}^{L} \text{CrossEntropy}(\hat{a}^{(0)}, a^{(0)}).$$

We also incorporate violation loss from AlphaFold (Jumper et al., 2021; Abramson et al., 2024) to learn the geometry for inter-residues bonds and avoidance of atom clashes. Specifically, these losses can be written as follows:

$$\mathcal{L}_{\text{bondlength}} = \frac{1}{N_{\text{bonds}}} \sum_{i=1}^{N_{\text{bonds}}} \max\left(|\ell_{\text{design}}^i - \ell_{lit}^i| - \tau_{\text{bondlength}}, 0\right),$$

$$\mathcal{L}_{\text{bondangle}} = \frac{1}{N_{\text{angles}}} \sum_{i=1}^{N_{\text{angles}}} \max\left(|\cos\alpha_{\text{design}}^i - \cos\alpha_{\text{lit}}^i| - \tau_{\text{bondangle}}, 0\right), \tag{32}$$

Where $\ell_{\text{design}}^i$ and $\cos\alpha_{\text{design}}^i$ are the bond length and bond angle of $i$-th designed antibodies, respectively. $\ell_{\text{lit}}^i$ and $\cos\alpha_{\text{lit}}^i$ is the literature value for this bond length and bond angle. We use same tolerance value ($\tau_{\text{bondlength}}$ and $\tau_{\text{bondangle}}$) as AlphaFold.

Additionally, we add auxiliary loss for training (Yim et al., 2023). $\mathcal{L}_{\text{aux}}$ is an mean squared error(MSE) loss supervising the distance of designed CDR's four atoms $\Omega \in \{C, C_\alpha, N, O\}$. This can be written as follows:

$$\mathcal{L}_{\text{aux}} = \frac{1}{4L} \sum_{i=1}^{L} \min\left(\sum_{a \in \Omega} ||x_{i,a}^{(t)} - \hat{x}_{i,a}^{(t)}||_2^2, d_{\text{clamp}}\right). \tag{33}$$

**Training Losses for Preference Optimization.** From Equation 9, $\mathcal{F}(\cdot)$ can be written as follows:

$$
\begin{aligned}
\mathcal{F}(\cdot) = & -\mathbb{D}_{\mathrm{KL}}\Big(q^{x,r,a}(\mathbf{T}_i^{(t-\Delta t)}|\mathbf{T}_i^{(0,t)})||p_\theta^{x,r,a}(\mathbf{T}_i^{(t-\Delta t)}|\mathbf{T}_i^{(t)})\Big) \\
& \qquad\qquad + \mathbb{D}_{\mathrm{KL}}\Big(q^{x,r,a}(\mathbf{T}_i^{(t-\Delta t)}|\mathbf{T}_i^{(0,t)})||p_{\mathrm{ref}}^{x,r,a}(\mathbf{T}_i^{(t-\Delta t)}|\mathbf{T}_i^{(t)})\Big) \\
= & -\alpha^x\Big[\mathbb{D}_{\mathrm{KL}}\Big(q^x(\mathbf{x}_i^{(t-\Delta t)}|\mathbf{x}_i^{(0,t)})||p_\theta^x(\mathbf{x}_i^{(t-\Delta t)}|\mathbf{x}_i^{(t)})\Big) \\
& \qquad\qquad - \mathbb{D}_{\mathrm{KL}}\Big(q^x(\mathbf{x}_i^{(t-\Delta t)}|\mathbf{x}_i^{(0,t)})||p_{\mathrm{ref}}^x(\mathbf{x}_i^{(t-\Delta t)}|\mathbf{x}_i^{(t)})\Big)\Big] \\
& -\alpha^r\Big[\mathbb{D}_{\mathrm{KL}}\Big(q^r(\mathbf{r}_i^{(t-\Delta t)}|\mathbf{r}_i^{(0,t)})||p_\theta^r(\mathbf{r}_i^{(t-\Delta t)}|\mathbf{r}_i^{(t)})\Big) \\
& \qquad\qquad - \mathbb{D}_{\mathrm{KL}}\Big(q^r(\mathbf{r}_i^{(t-\Delta t)}|\mathbf{r}_i^{(0,t)})||p_{\mathrm{ref}}^r(\mathbf{r}_i^{(t-\Delta t)}|\mathbf{r}_i^{(t)})\Big)\Big] \\
& -\alpha^a\Big[\mathbb{D}_{\mathrm{KL}}\Big(q^a(\mathbf{a}_i^{(t-\Delta t)}|\mathbf{a}_i^{(0,t)})||p_\theta^a(\mathbf{a}_i^{(t-\Delta t)}|\mathbf{a}_i^{(t)})\Big) \\
& \qquad\qquad - \mathbb{D}_{\mathrm{KL}}\Big(q^a(\mathbf{a}_i^{(t-\Delta t)}|\mathbf{a}_i^{(0,t)})||p_{\mathrm{ref}}^a(\mathbf{a}_i^{(t-\Delta t)}|\mathbf{a}_i^{(t)})\Big)\Big],
\end{aligned}
\tag{34}
$$

where $\mathbf{x}_i = [x_{n+1,i}, ..., x_{n+m,i}]$, $\mathbf{r}_i = [r_{n+1,i}, ..., r_{n+m,i}]$, and $\mathbf{a}_i = [a_{n+1,i}, ..., a_{n+m,i}]$, the first subscript represents the $i$-th sample in the preference optimization and the second subscript represents the $j$-th amino acid in the CDR. With some algebra, these KL divergences can be derived as the following form:

**KL Divergence in $\mathbb{R}^3$ Space.** According to Wallace et al. (2024), the closed-form expressions of KL divergence in $\mathbb{R}^3$ simplifies to:

$$
\mathbb{D}_{\mathrm{KL}}^{(x)} = ||\mathbf{f}_\theta^x(\mathbf{x}^{(t)}) - \mathbf{x}^{(0)}||_2^2 + C.
\tag{35}
$$

**KL Divergence in SO(3) Space.** Similar to Zhou et al. (2024), we approximately derive an empirical reconstruction loss in SO(3) as:

$$
\mathbb{D}_{\mathrm{KL}}^{(r)} = ||\nabla_{\mathbf{r}}\log p_{t|0}(\mathbf{r}^{(t)}|\mathbf{f}_\theta^r(\mathbf{r}^{(t)})) - \nabla_{\mathbf{r}} p_{t|0}(\mathbf{r}^{(t)}|\mathbf{r}^{(0)})||_2^2 + C.
\tag{36}
$$

**KL Divergence in Discrete Space.** For sequences in discrete space, we give the details of the derivation process:

$$
\begin{aligned}
\mathbb{D}_{\mathrm{KL}}^{(a)} &= \mathbb{D}_{\mathrm{KL}}\Big(q^a(\mathbf{a}^{(t-\Delta t)}|\mathbf{a}^{(0)}, \mathbf{a}^{(t)})||p_\theta^a(\mathbf{a}^{(t-\Delta t)}|\mathbf{a}^{(t)})\Big) \\
&= \mathbb{E}_{q^a(\mathbf{a}^{(t-\Delta t)}|\mathbf{a}^{(0)},\mathbf{a}^{(t)})}\Big[-\log p_\theta^a(\mathbf{a}^{(t-\Delta t)}|\mathbf{a}^{(t)})\Big] + C_1 \\
&= \mathbb{E}_{q^a(\mathbf{a}^{(t-\Delta t)}|\mathbf{a}^{(0)},\mathbf{a}^{(t)})}\Big[-\log\sum_{\tilde{\mathbf{a}}^{(0)}} q^a(\mathbf{a}^{(t-\Delta t)}|\tilde{\mathbf{a}}^{(0)},\mathbf{a}^{(t)})p_\theta^a(\tilde{\mathbf{a}}^{(0)}|\mathbf{a}^{(t)})\Big] + C_1 \\
&= \mathbb{E}_{q^a(\mathbf{a}^{(t-\Delta t)}|\mathbf{a}^{(0)},\mathbf{a}^{(t)})}\Big[-\log\sum_{\tilde{\mathbf{a}}^{(0)}} \frac{q^a(\tilde{\mathbf{a}}^{(0)}|\mathbf{a}^{(t-\Delta t)})q^a(\mathbf{a}^{(t-\Delta t)}|\mathbf{a}^{(t)})}{q^a(\tilde{\mathbf{a}}^{(0)}|\mathbf{a}^{(t)})}p_\theta^a(\tilde{\mathbf{a}}^{(0)}|\mathbf{a}^{(t)})\Big] + C_1 \\
&\leq \mathbb{E}_{q^a(\mathbf{a}^{(t-\Delta t)}|\mathbf{a}^{(0)},\mathbf{a}^{(t)}),q^a(\tilde{\mathbf{a}}^{(0)}|\mathbf{a}^{(t-\Delta t)})}\Big[-\log\frac{q^a(\mathbf{a}^{(t-\Delta t)}|\mathbf{a}^{(t)})}{q^a(\tilde{\mathbf{a}}^{(0)}|\mathbf{a}^{(t)})}p_\theta^a(\tilde{\mathbf{a}}^{(0)}|\mathbf{a}^{(t)})\Big] + C_1 \\
&= \mathbb{E}_{q^a(\tilde{\mathbf{a}}^{(0)},\mathbf{a}^{(t-\Delta t)}|\mathbf{a}^{(0)},\mathbf{a}^{(t)})}\Big[-\log p_\theta^a(\tilde{\mathbf{a}}^{(0)}|\mathbf{a}^{(t)})\Big] \\
&\quad + \mathbb{E}_{q^a(\tilde{\mathbf{a}}^{(0)},\mathbf{a}^{(t-\Delta t)}|\mathbf{a}^{(0)},\mathbf{a}^{(t)})}\Big[-\log\frac{q^a(\mathbf{a}^{(t-\Delta t)}|\mathbf{a}^{(t)})}{q^a(\tilde{\mathbf{a}}^{(0)}|\mathbf{a}^{(t)})}\Big] + C_1 \\
&= -\log p_\theta^a(\mathbf{a}^{(0)}|\mathbf{a}^{(t)}) + C.
\end{aligned}
\tag{37}
$$

In practice, we choose $[\alpha^{(x)}, \alpha^{(r)}, \alpha^{(a)}]=[1.0, 0.5, 0.2]$, $\alpha^{(\mathrm{sup})} = 0.5$ and $K = 8$. Meanwhile, we add the regularisation term $\alpha^{(\mathrm{R})}$ for $\frac{1}{\Delta t}$ in Equation 9 to ensure the stability of training. Here, we choose $\alpha^{(\mathrm{R})}/\Delta t = 10.0$.

### A.4.6 TRAINING DETAILS

We show information about the training process, objectives, and learning rate of AbNovo in Table 14.

Especially in the fine-tuning stage, for updating $\lambda$ for one step, we update the policy for 100 steps. We show details about losses we used when updating policy and $\lambda$ in 3.3.1 and 3.3.2.

We use 8 Nvidia A100 (80G) for training, and the batch size is 128 for all training stages. For all training procedures, we use the Adam optimizer for training with default parameters.

Table 14: Hyper-parameter of AbNovo.

| Stage | Training objective | Training steps | Learning Rate | Dataset |
|---|---|---|---|---|
| Pre-trained | $\mathcal{L}_{\text{MLM}} + \mathcal{L}_{\text{distogram}} + \mathcal{L}_{\text{contact}}$ | 200k | 5e-5 | AFDB (2M)+PDB(filter) |
| Base model | $1.0\mathcal{L}^{(x)} + 0.5\mathcal{L}^{(r)} + 0.2\mathcal{L}^{(a)} +$ $0.1\mathcal{L}_{\text{violation}} + 1.0\mathcal{L}_{\text{aux}}$ | 20k | 1e-4 | Antigen-antibody complex |
| Fine-tuning | $\mathcal{L}_{\text{update policy}}$ (Equation 10) | 20k | 2e-5 | Preference dataset |

### A.5 EVALUATION METRICS

In this section, we provide detailed descriptions of each metric.

**Rosetta Binding Energy.** For Rosetta Binding Energy, we followed previous work (Zhou et al., 2024) by calculating the energy between the antigen and antibody using Rosetta Software. Lower energy values represent higher antigen-antibody affinity.

We first used Rosetta software to perform side-chain packing on the backbone and side chains of the designed antibodies, followed by *FastRelax* on the side chains. Then, we calculated the Rosetta Binding Energy. We denote the residue with the index $i$ in the antibody-antigen complex as $A_i$, then $A_i^{\text{sc}}$ and $A_i^{\text{bb}}$ represent the side chain and backbone of the residue respectively. We use EP to represent the interaction energies between Paired residues, which consists of six different energy types: $E_{\text{hbond}}, E_{\text{att}}, E_{\text{sol}}, E_{\text{elec}}, E_{\text{lk}}, E_{\text{rep}}$.

$$
\begin{aligned}
E = \sum_{j\in\text{CDRs}} \sum_{i\in\text{Antigens}} & \Big( \text{EP}_{\text{rep}}(A_j^{\text{sc}}, A_j^{\text{sc}}) + \text{EP}_{\text{rep}}(A_j^{\text{sc}}, A_j^{\text{bb}}) \\
& + 2\text{EP}_{\text{rep}}(A_j^{\text{bb}}, A_j^{\text{sc}}) + 2\text{EP}_{\text{rep}}(A_j^{\text{bb}}, A_j^{\text{bb}}) \Big) \\
& + \sum_{j\in\text{CDRs}} \sum_{i\in\text{Antigens}} \sum_{e\in\{\text{hbond,att,elec,lk,rep}\}} \Big( \text{EP}_{\text{e}}(A_j^{\text{sc}}, A_j^{\text{sc}}), \text{EP}_{\text{e}}(A_j^{\text{sc}}, A_j^{\text{bb}}) \Big)
\end{aligned}
\tag{38}
$$

The calculation incorporates various energy terms from Rosetta, using the default weights in *ref2015* for each term.

**Evolutionary Plausibility.** Evolutionary Plausibility measures how likely a designed sequence is evolutionarily plausible in nature, reflecting adherence to general evolutionary rules of natural proteins. Recent studies show that large-scale protein language models, trained on millions of natural protein sequences, effectively capture these evolutionary rules (Shuai et al., 2021; Hie et al., 2024). Following previous works (Zhu et al., 2024; Zhou et al., 2024), we evaluate Evolutionary Plausibility by calculating the perplexity of the antibody language model (Shuai et al., 2021) for the designed antibodies. Here, we use $a_i$ to represent the residue type of designed CDRs. $P$ represents the conditional probability of BERT predicting the word at masked position $i$, where $\theta$ represents the model parameters. The specific formula is as follows:

$$
\text{Evolutionary Plausibility} = - \sum_{i\in\text{CDRs}}^{N} \log P(a_i | a_i, ..., a_{i-1}, a_{i+1}, ..., a_N)
$$

We used the script provided by IgLM Shuai et al. (2021) to calculate this metric.

**Stability.** Stability measures the stability of the conformation of the designed antibody in isolation, without the antigen structure involved. This metric differs from Binding Energy, which evaluates the interaction between the antibody and the antigen. Following the evaluation approach of prior work (Zhou et al., 2024; Li et al., 2024), we used Rosetta to calculate the total energy of the designed antibody to assess its stability.

We performed side-chain packing and FastRelax on the backbone and side chains of the designed antibody. Then we output the total energy of the CDR regions using the Rosetta software.

**Self-association.** The self-association of antibodies refers to the tendency of antibody molecules to bind to each other without interacting with the antigen. Self-association can negatively impact the stability, function, and biological properties of antibodies, especially in the context of therapeutic antibody development, where self-aggregation is generally undesirable (Yadav et al., 2012; Kanai et al., 2008).

Previous studies (Makowski et al., 2024) have shown that the occurrence of self-association is closely correlated with the negatively charged patches area in CDRs. Therefore, we use negatively charged patch areas in CDRs as a proxy for evaluating the risk of self-association. Although other physicochemical properties are also correlated with self-association, we used the physicochemical property of CDRs with the highest correlation as the evaluation metric. We used the pipeline provided by previous work (Makowski et al., 2024) (`https://github.com/Tessier-Lab-UMich/CST_Prop_Opt_ML`) to calculate the self-association metric.

**Non-specific Binding.** Non-specific binding refers to the undesirable interaction of antibodies with cellular proteins other than the intended target, particularly membrane proteins of the cell (Makowski et al., 2022). In practice, evaluating non-specific binding would require consideration of interactions with all membrane proteins and RNA-specific binding properties. Computationally modeling these interactions for all potential targets is a highly complex and challenging task. Here, we followed the metric proposed in previous works (Makowski et al., 2024; 2022).

Previous studies (Makowski et al., 2024; 2022) have verified a correlation between non-specific binding and the hydrophobic patches area in CDRs. Therefore, we use hydrophobic patch areas in CDRs as a proxy for evaluating the risk of non-specific binding. Although other physicochemical properties also have correlation with non-specific binding, we used the most strongly correlated physicochemical property in CDRs as the evaluation metric. We used the pipeline provided by previous work (Makowski et al., 2024) (`https://github.com/Tessier-Lab-UMich/CST_Prop_Opt_ML`) to calculate the non-specific binding metric.

**RMSD.** For DiffAb, AbX, GeoAb, and AbNovo, where the antibody framework structure is provided, we directly calculate the RMSD of the designed antibody. However, for dyMEAN, as there is no given real antibody framework, we use Kabsch alignment to align the designed antigen-antibody complex with the natural complex. Then, we calculate the Root Mean Square Deviation (RMSD) for each region of these aligned complexes.

**Reward and Constraints Settings.** Given the stringent requirements for compliance with clinical applications, as well as the inherent biases in hydrophobicity and polarity metrics, our primary objective is to filter out antibodies that clearly violate binding rules while retaining as many potential candidates as possible. Setting an excessively strict threshold would, on the one hand, eliminate many viable candidate antibodies and, on the other hand, destabilize the model's training.

Consequently, we heuristically determined the threshold for each constraint based on training stability and the empirical thresholds used in previous studies.

Specifically, we determine the thresholds based on the dataset provided in previous work (Makowski et al., 2024). This dataset comprises 80 clinical antibodies annotated with corresponding *wet-lab* experimental results. For non-specific binding and self-association metrics, we calculate the physicochemical properties of all antibodies that meet the requirements, setting the threshold for constraint violation as twice the highest value of the physicochemical property observed in the antibody that satisfies the *wet-lab* metric.

For stability assessment, we evaluated all antibodies in RAbD and set the highest energy value as the threshold for constraint violation.

In our experiments, considering the different magnitudes of various rewards and constraints, we normalized all rewards and constraints during the training process. Meanwhile, we set the initial $\boldsymbol{\lambda}$ to $[1.0, 1.0, 1.0]$ and the reward weights $\omega_1$ and $\omega_2$ to 1.0 and 1.0.

## A.6    BASELINE METHODS

**DiffAb** (Luo et al., 2022).    We use the test script and the pre-trained model provided in the GitHub repository (`https://github.com/luost26/diffab/tree/main`). All hyper-parameters are default.

**dyMEAN** (Kong et al., 2023).    We use the pre-trained model and test script provided in the GitHub repository (`https://github.com/THUNLP-MT/dyMEAN`). All hyper-parameters we used are default.

**GeoAb** (Lin et al., 2024).    We employed GeoAb from its GitHub repository (`https://github.com/Edapinenut/GeoAB`) with all default hyper-parameters and the pre-trained model provided.

**AbX** (Zhu et al., 2024).    We use the co-design test script and pre-trained model provided in the GitHub repository (`https://github.com/CarbonMatrixLab/AbX`). All hyper-parameters we used are default.

**AbDiffuser** (Martinkus et al., 2024)**, AbDPO** (Zhou et al., 2024)**, and AbDesigner** (Tan et al., 2024).    These models were not available at the time of this paper's submission. Moreover, the results presented in their papers are not compatible with our experiment settings. For example, AbDPO only designs CDR-H3; AbDiffuser does not provide six CDRs designed results simultaneously; and AbDesigner uses a different test set. Thus, we do not include their results in our paper.

