# OpenReview forum: "Multi-objective antibody design with constrained preference optimization"
_ICLR.cc/2025/Conference — ICLR 2025 Poster_

### Official Review · Reviewer_fXjU · 2024-10-29

**Soundness:** 2
**Presentation:** 2
**Contribution:** 3
**Rating:** 8
**Confidence:** 3

**Summary:**

In this manuscript, the authors present AbNovo, a method combining constrained preference optimization with generative models for multi-objective antibody design. First, an antigen-conditioned generative model is trained to co-design antibody structure and sequence. Then this model is fine-tuned to maximize binding affinity to a target antigen while enforcing constraints on properties such as non-specific Binding, Self-association, and Stability. In their experiments, the authors compare their method to many recent works and show an improved performance.

**Strengths:**

•	The authors provide many theoretical derivations and analysis.

•	The authors include many baselines for their experiments which shows the good performance of their proposed method.

**Weaknesses:**

•	I would suggest introducing a background section, as there are many things in this manuscript that would benefit from a proper introduction.

  	∘	base model, reference model, policy model could be introduced, e.g. with an intuition. These are introduced in Figure 1, but do not come with a description on how they are related. Only in Algorithm 1, the reader is shown that those are updated iterations of the very same model.

  	∘	delta and G in Equation 1 is never introduced, but instead taken from Campbell et al.

  	∘	CTMC - Continuous Time Markov Chain is never defined.

  	∘	The notion of time t in diffusion processes used in the manuscript, t in U([0, 1]) is based on the CTMC definition by Campbell et al. but differs to that used on many other publications, e.g. [1] J. Ho, A. Jain, and P. Abbeel, “Denoising Diffusion Probabilistic Models”, [2] J. Sohl-Dickstein, E. Weiss, N. Maheswaranathan, and S. Ganguli, “Deep Unsupervised Learning using Nonequilibrium Thermodynamics,”. Thus, I would recommend introducing it e.g. as being in [0, 1] in line 185. Instead, this is first done in Equation 9.

  	∘	T^(0:1) as a diffusion path is first defined in line 284 even tough being used many times before.


•	The evaluation metrics remain unclear even after reading the appendix A.3. This holds especially for “Evolutionary Plausibility”, “Stability”, “Self-association”, “Non-specific Binding”.

•	While the manuscript goes in great theoretical detail, intuition is often lacking. E.g. Equation 3 is introduced but an intuition, “first term maximizes rewards, while second term keeps the model close to the reference model.”, which could facilitate understanding for the reader is missing.

•	Many things necessary for fully understanding the paper are moved to the appendix, resulting in decreased readability. Further, this also applies to some of the most interesting results, e.g. Table 9 and especially Figure 4.

•	Some tables are hard to read, as their caption and corresponding text do not exactly describe what is in the table. E.g.

  	∘	It is unclear what “reference“ in Table 1 describes.
  	∘	In Table 3 the reader must guess that “ESM-2 based” refers to “utilizing different language models” from the text and “Multi-objective” refers to “we incorporated all constraints into the optimization objective”.

•	In the abstract and introduction, a focus is put on “alleviate overfitting issues due to the scarcity of antibody-antigen training data”, but no analysis supporting such a claim is included.

•	The analysis of the “impact of utilizing different language models in training the antibody design model” is very short and not well described.

•	Figure 4 is a very interesting figure which summarizes the capabilities of DiffAb, AbX, and AbNovo very well and highlights that AbNovo “performs best“. In there, we also observe that only a single antibody generated by DiffAb against 5NUZ does violate constraints. Therefore, it seems inadequate that the visualized antibody for DiffAb in Figure 2 is a sample which does not fulfill all constraints. Furthermore, the DiffAb sample with “Rosetta binding energy: -2.12, Evolutionary Plausibility: 2.60” violating constraints cannot be found in Figure 4.

•	Some claims appear exaggerated:

  	∘	“the first deep generative model for multi-objective antibody design, which explicitly optimizes multiple biophysical properties crucial for real-world antibody development.” There have been previous works which analyze the multi-objective setting for generating antibodies, e.g. “Pareto Front Training For Multi-Objective Symbolic Optimization" by Faris et al. which train a algorithm to optimize a pareto front of sequences regarding the objectives antibody binding quality, stability, and humanness. Perhaps the claim can be weakened or reformulated?

  	∘	AbNovo is “bridging the gap between in silico design and practical application.” seems a bit too strong given that no practical application is contained.

•	Typo “Bolocks” in Figure 3

In summary, I think this manuscript offers valuable new ideas but suffers from not being self-contained, sub-optimal readabilities and depth of analysis. I hope these issues can be addressed in the rebuttal and would love to increase my score in response.

**Questions:**

•	in Section 4.2 you state that when “we incorporated all constraints into the optimization objective by taking a weighted average” a “drop in performance” is observable. However, the corresponding results show an improvement wrt. the “All Constraints” metric. Could you elaborate on that?

•	In Table 2, we can observe that AbNovo (base) sometimes exhibits favorable scores than AbNovo. Is there a tradeoff between fulfilling constraints and achieved AAR/RMSD?

•	Is there a reason dyMEAN is not included in Figure 4 and AbX not in Figure 2 respectively?

-------
Post rebuttal:
All points were well addressed. Based on this, I am increasing my score accordingly.

---

> ### Author Response · Authors · 2024-11-21
>
> Thank you for your valuable suggestions, which have significantly improved the quality of our manuscript. We have provided point-by-point responses to your comments below. We hope that our revisions and additional experiments address your concerns satisfactorily.
>
> **Weakness 1: I would suggest introducing a background section, as there are many things in this manuscript that would benefit from a proper introduction.**
>
> We added more background introduction on both preference optimization and the diffusion-based generative model.  For preference optimization, we introduce concepts on the beginning of our method (Line 159-163).  We introduce more details on Equation 1 (Line 202-204), the definition of CTMC (Line 186), and $T^{(0:1)}$ (Line 191).  Additionally, we explicitly describe that the time $t$ in diffusion processes is a uniform distribution of [0, 1] (Line 191-192).
>
> To further enhance the readability of our manuscript, we have provided a detailed notation table for all symbols used throughout the paper, which can be found in Appendix A.1.
>
> **Weakness 2: The evaluation metrics remain unclear even after reading the appendix A.3. This holds especially for “Evolutionary Plausibility”, “Stability”, “Self-association”, “Non-specific Binding”.**
>
> We have revised the this part to make these terms  more clear (Line 1427-1497) and show how the different metrics are calculated and what they represent. We also briefly explain these terms below.
>
> **Evolutionary Plausibility:**
>
> *Evolutionary Plausibility* measures how likely a designed sequence  is evolutionarily plausible in nature, reflecting adherence to general evolutionary rules of natural proteins. Recent studies show that large-scale protein language models, trained on millions of natural protein sequences, effectively capture these evolutionary rules [1, 2].  Specifically, we calculate this metric as the log-likelihood of the designed sequence under a pre-trained protein language model (Line 1457). Importantly, this approach has proven useful for guiding antibody maturation in wet-lab experiments [1], and it is thus widely used as an evaluation metric in recent generative models for antibody design [3, 4, 5].
>
> **Stability:**
>
> *Stability* measures the stability of the conformation of designed antibody in isolation, without the antigen structure involved. This metric differs from *Binding Energy*, which evaluates the interaction between the antibody and the antigen. Specifically, we calculate this metric using established protocols based on the Rosetta software, as employed in previous methods [6].
>
> **Self-association:**
>
> *Self-Association* refers to the tendency of antibody molecules to aggregate with each other. Self-association can negatively impact the efficacy of antibodies, so low self-association is desired in practical antibody development [7, 8]. Previous studies have shown that larger negatively charged patches area in the CDRs corresponds to a higher risk of self-association [7]. Specifically, we calculate this metric using established protocols from previous methods [7].
>
> **Non-specific Binding:**
>
> *Non-specific Binding*, or *Binding Specificity*, refers to the undesirable interaction of antibodies with cellular proteins other than the intended target, particularly membrane proteins of the cell. Previous studies [7] have observed a strong correlation between non-specific binding and the hydrophobic patches in CDRs. Therefore, we use the hydrophobic patch area in CDRs as a proxy for evaluating the risk of non-specific binding, utilizing their established pipeline to calculate this metric [7].
>
> [1] Hie, et al. [Efficient evolution of human antibodies from general protein language models.](https://www.nature.com/articles/s41587-023-01763-2) Nature Biotechnology 2024.
>
> [2] Shuai, et al. [IgLM: Infilling language modeling for antibody sequence design.](https://www.cell.com/cell-systems/fulltext/S2405-4712(23)00271-5) Cell System 2023.
>
> [3] Zhu, et al. [Antibody Design Using a Score-based Diffusion Model Guided by Evolutionary.](https://openreview.net/pdf?id=1YsQI04KaN) ICML 2024.
>
> [4] Kong, et al. [End-to-End Full-Atom Antibody Design](https://arxiv.org/abs/2302.00203). ICML 2023.
>
> [5] Zhou, et al. [Antigen-Specific Antibody Design via Direct Energy-based Preference Optimization.](https://arxiv.org/pdf/2403.16576v1) NeurIPS 2024.
>
> [6] Li, et al. [Full-atom peptide design based on multi-modal flow matching.](https://arxiv.org/abs/2406.00735) ICML 2024.
>
> [7] Makowski, et al. [Optimization of therapeutic antibodies for reduced self-association and non-specific binding via interpretable machine learning.](https://www.nature.com/articles/s41551-023-01074-6) Nature Biomedical Engineering 2023.
>
> [8] Yadav, et al. [The influence of charge distribution on self-association and viscosity behavior of monoclonal antibody solutions.](https://pubs.acs.org/doi/abs/10.1021/mp200566k) Molecular pharmaceutics 2012.

---

> ### Author Response · Authors · 2024-11-21
>
> **Weakness 3: While the manuscript goes in great theoretical detail, intuition is often lacking. E.g. Equation 3 is introduced but an intuition, “first term maximizes rewards, while second term keeps the model close to the reference model.”, which could facilitate understanding for the reader is missing.**
>
> This suggestion is great to enhance readability of our manuscript. We have provided intuitive explanations for the key formulas in the article.  For example,
>
> 1. We provided intuitive explanations for concepts such as the primal-dual algorithm and the dual gradient estimation (Equation 4,5). This equation can be interpreted as appending a penalty term ${J}^{({C})}$ to the original objective ${J}^{({R})}$. The penalty term, which depends on how much the antibody generated by the current model violates constraints, can be adjusted dynamically through the Lagrange multipliers. (Line 256-258).
> 2. We also provided intuitive explanations for the gradient of Lagrange multipliers (Equation 11). In this equation, the gradient of dual can be calculated by the expected degree of constraint violation in the sampled antibodies under the current policy model. (Line 332-334)
>
> Intuitive description for more formulas can be found in Line 244-246 and 300-302.
>
> **Weakness 4: Many things necessary for fully understanding the paper are moved to the appendix, resulting in decreased readability. Further, this also applies to some of the most interesting results, e.g. Table 9 and especially Figure 4.**
>
> Following your suggestions, we have moved some of the important results (Table 9) to the main text.  However, due to the strict 10-page limit for the main text at the ICLR conference, we were unable to present more results in the main paper. Therefore, some results were included in the supplementary materials, with a clear link provided in the main text. We placed the important experimental results at the beginning of the Appendix.
>
> **Weakness 5: Some tables are hard to read, as their caption and corresponding text do not exactly describe what is in the table.**
>
> Following your suggestions, we have revised the captions and provided detailed explanations for better understanding. Specifically:
>
> In Table 1, "reference" represents the native antibody structure and sequence in the testing set.
> (Line 420-421)
>
> For Table 3, we have explicitly clarified the meaning of “ESM-2 based” and “Multi-objective.” Below is the updated caption:
>
> Ablation studies for AbNovo on the RAbD dataset. The ablation experiment settings include: without using a language model (w.o. LM), replacing the structure-aware language model with ESM2 (ESM-2 based), using supervised fine-tuning instead of preference optimization (SFT), and incorporating all constraints into the optimization objectives (Multi-objective).
>
> Similarly, we revised other unclear parts of the article to improve clarity (Line 153-156, 461-466 , 486-488, and 518-519).
>
> **Weakness 6: In the abstract and introduction, a focus is put on “alleviate overfitting issues due to the scarcity of antibody-antigen training data”, but no analysis supporting such a claim is included.**
>
> To alleviate the scarcity of antibody-antigen training data, we utilized  a structure-aware  language models pre-trained on massive structures beyond antibody proteins.  We included two ablation experiments to demonstrate the relative contribution of the structure-aware language model. (Line 447-456)
>
> First, we trained an ablation model where we exclude the embeddings of the language model as input features.  We observed significant drops across nearly all metrics, indicating the importance of the language model.
>
> Second, we also trained an ablation model  where we replace this structure-aware language model with a sequence-only language model (ESM2). It shows that the structure-aware model yielded better results than  the purely sequence-based model.
>
> |  | Rosetta Binding Energy | Evolutionary Plausibility | Constraints | AAR | RMSD |
> | --- | --- | --- | --- | --- | --- |
> | w.o. language model | 7.54 | 2.67 | 46.5% | 41.53% | 3.19 |
> | ESM2 | 1.75 | 2.40 | 30.8% | 49.2% | 2.55 |
> | AbNovo (base) | -2.60 | 2.41 | 22.7% | 49.9% | 2.19 |
>
> **Weakness 7: Question about case studies**
>
> We have now utilized another more reasonable case where the DiffAb fulfill all constraints.  We note that though the antibody generated by DiffAb fulfilled all constraints, AbNovo outperformed DiffAb in terms of Rosetta Binding Energy and Evolutionary Plausibility.
>
> We also presented more details in the updated figure, such as the CDR H3 sequences  annotated with biophysical properties.

---

> > ### Author Response · Authors · 2024-11-21
> >
> > **Weakness 8: Some claims appear exaggerated**
> >
> > We have addressed these concerns in the revised manuscript. Specifically:
> >
> > 1. We have removed the word “first” from the sentence (Line 71-73).
> > 2. We have removed the claim that AbNovo is “bridging the gap between in silico design and practical application” from the paper (Line 81-82).
> >
> > **Weakness9: Typo “Bolocks” in Figure 3**
> >
> > Thanks for pointing out this typo. We have fixed it.
> >
> > **Question 1: In Section 4.2 you state that when “we incorporated all constraints into the optimization objective by taking a weighted average” a “drop in performance” is observable. However, the corresponding results show an improvement wrt. the “All Constraints” metric. Could you elaborate on that?**
> >
> > For a more accurate description, we have updated the statement in the manuscript to: “We observed a slight increase in fulfilling all constraints but a significant drop in performance in Binding Energy and Evolutionary Plausibility.” (Line 442-444).
> >
> > When we incorporate all constraints into the optimization objective, the model  struggles to balance these objectives effectively, often allowing one objective to dominate at the expense of others.  While this ablation model achieves slightly better performance in fulfilling all constraints, it performs significantly worse in metrics such as Rosetta Binding Energy and Evolutionary Plausibility.
> >
> > A similar observation has been made in language model preference optimization, where a constrained preference optimization framework is shown to be more effective than a purely preference optimization framework in balancing user-helpful responses against safety concerns (e.g., avoiding offensive content) [1, 2].
> >
> > [1] Bianchi,et al. [Safety-tuned llamas: Lessons from improving the safety of large language models that follow instructions](https://arxiv.org/abs/2309.07875). ICLR 2023.
> >
> > [2] Liu, et al. [Enhancing LLM Safety via Constrained Direct Preference Optimization.](https://arxiv.org/abs/2403.02475) ICLR WorkShop 2024.
> >
> > **Question 2: In Table 2, we can observe that AbNovo (base) sometimes exhibits favorable scores than AbNovo. Is there a tradeoff between fulfilling constraints and achieved AAR/RMSD?**
> >
> > Yes, as you observed, there is a slight trade-off between fulfilling constraints and achieving AAR/RMSD, which is consistent with findings in previous work [1]. In the base model, the training objectives focus on generating antibodies that closely align with native antibodies in terms of sequence and structure. After applying preference optimization, we found that a slight sacrifice in AAR and RMSD can lead to significant improvements in other metrics, such as binding affinity, evolutionary plausibility, and other important biochemical properties..
> >
> > [1] Zhou, et al. [Antigen-Specific Antibody Design via Direct Energy-based Preference Optimization.](https://arxiv.org/pdf/2403.16576v1) NeurIPS 2024.
> >
> > **Question 3: Is there a reason dyMEAN is not included in Figure 4 and AbX not in Figure 2 respectively?**
> >
> > Figure 4 illustrates the distribution of designed antibodies  for one specific antigen. Since dyMEAN cannot generate diverse antibodies for a given antigen, it is excluded from Figure 4.
> >
> > We have now included AbX in Figure 2. The main reason we omitted AbX in the previous version was that including it would have made the figure look too busy.
> >
> > Thank you for your constructive feedback on our paper. If you have any further concerns, please feel free to discuss them with us. We look forward to your response.

---

> > > ### Comment · Reviewer_fXjU · 2024-11-24
> > >
> > > Thank you for your detailed feedback. All points were well addressed, and I appreciate the clarity of your responses. Based on this, I am increasing my score accordingly.

---

> > > > ### Author Response · Authors · 2024-11-25
> > > >
> > > > Once again, we sincerely appreciate the reviewer's insightful and constructive comments, which have significantly enhanced the quality of our manuscript.

---

### Official Review · Reviewer_vnHk · 2024-10-30

**Soundness:** 3
**Presentation:** 3
**Contribution:** 3
**Rating:** 6
**Confidence:** 4

**Summary:**

This paper presents an antibody design method, AbNovo, achieved antibody through multi-objective optimization. By introducing a structure-aware protein language model and employing constrained preference optimization with continuous rewards, AbNovo surpasses previous methods in both reference-based metrics and reference-free metrics (i.e., biophysical properties).

**Strengths:**

Achieved performance on physical metrics that significantly surpasses other methods.

Introduced a structure-aware protein language model and demonstrated its usefulness for antibody design.

Provided rigorous theoretical derivation

**Weaknesses:**

Seems to be an updated version of AbDPO, somewhat heavier but showing better performance.

The task setting is overly simplistic. Although the structure of the antibody's FR region is relatively conserved and can be considered known, the binding pose between the antibody and antigen is typically unknown. However, given that the main goal of this work is to propose a new method for antibody optimization, this limitation is understandable.

**Questions:**

1. The announcement of "The first deep generative model for multi-objective antibody design" in summarized contributions, AbDPO also supports multi-objective optimization.

2. In energy evaluation, if you want to assess the energy performance of the designed backbone, energy minimization is necessary for the side chains while keeping the backbone structure unchanged, and then calculate the energy. If you wish to evaluate the antibody's performance in real experiments (which implies the CDR region's structure might not maintain the designed configuration), you can use multi-chain supporting folding models like AlphaFold3 to predict the binding structure. When calculating energy, does the relaxation you used optimize only the side chain conformations, or does it also alter the main chain structure? If it's the latter, are these experiments intended to demonstrate that AbNovo can generate a better initial structure for Rosetta relaxation?

3. Does the optimization of these physical properties contribute to some chemical validity? For example, does the peptide bond length get closer to the actual length?

4. The standard deviation of the physical energy needs to be presented.

5. The AAR performance is excessively high, and it's necessary to check whether the training data of the protein language model contains samples similar to the test set.

6. I am curious about how many amino acids have mutated in those designed antibodies that outperform natural ones (at least in binding energy).

7. The task setting of dyMEAN is different from others, including AbNovo. dyMEAN does not provide the real FR structure, making direct comparison somewhat unfair. Additionally, how is it achieved to use dyMEAN to generate 128 antibodies for an antigen?

8. Calculating RMSD on the aligned structures seems somewhat unreasonable. Typically, for two rigid bodies that can freely undergo SE(3) transformations, alignment is performed first, followed by RMSD calculation. However, in the setting of this paper, the FR region is given, meaning the CDR region cannot undergo SE(3) transformations independently, thus requiring a direct RMSD calculation.

---

> ### Author Response · Authors · 2024-11-21
>
> We sincerely appreciate your valuable suggestions, which have helped us further improve the quality of our article. We have provided point-to-point responses  to your comments as follows.
>
> **Weakness 1: Seems to be an updated version of AbDPO, somewhat heavier but showing better performance.**
>
> Our method differs with AbDPO in both the optimization framework and the underlying motivation.
>
> **AbDPO** employs direct preference optimization:
>
> $\max \ \ \ \ \sum_{i} R_i-\beta \mathrm{KL}(p_{\theta} |p_{\rm ref}) \ \ \ \ \$Equation 1.
>
> In contrast, **AbNovo** utilizes constrained preference optimization:
>
> $\max  \ \ \ \ \ \ \sum_{i} R_i-\beta \mathrm{KL}(p_{\theta} |p_{\rm ref}) \ \ \ \ $Equation  2.
>
> ${\rm s.t.} \ \ \ \ C_j<C_{{\rm limit},j} \ \ \ \ \$  for $j$ in all constraint sets
>
> Our framework is novel in diffusion-based generative models and is supported by rigorous theoretical derivations and proofs.
>
> The key motivation is that crucial biochemical properties in practical antibody development often present inherent trade-offs—for example, improving binding affinity may increase the risk of non-specific binding to non-target proteins [1, 2].
> Simply combining multiple objectives, as in Equation 1, struggles to balance these trade-offs effectively, often allowing one objective to dominate at the expense of others.  In contrast, the constrained preference optimization framework allows us to set thresholds for certain properties (e.g., specificity) while optimizing others (e.g., affinity).  It enables dynamic adjustment of the relationship between 'objectives' and 'constraints' through the Lagrangian method, thereby mitigating trade-offs to some extent.
>
> In our initial submission, we included an ablation study comparing the two optimization frameworks (see the "Multi-objective" row in Table 2, Line 474). In this study, we incorporated all constraints into the objective function as shown in Equation 1. The results demonstrate that AbNovo achieves better performance in Rosetta Binding Energy, Evolutionary Plausibility, RMSD, and AAR. In contrast, the ablation model tends to focus solely on satisfying the constraints, resulting in slight benefits in the metric of fulfilling all constraints but substantially sacrificing performance in other metrics.
>
> Similar trade-offs are observed in language model preference optimization, balancing user-helpful responses against safety concerns (e.g., avoiding offensive content) [3, 4].  One of our contributions is extending the constrained optimization framework for language models to multimodal diffusion models, providing theoretical support for this extension.
>
> [1]  Makowski, et al. [Optimization of therapeutic antibodies for reduced self-association and non-specific binding via interpretable machine learning.](https://www.nature.com/articles/s41551-023-01074-6) Nature Biomedical Engineering 2023.
>
> [2] Makowski, et al. [Co-optimization of therapeutic antibody affinity and specificity using machine learning models that generalize to novel mutational space.](https://www.nature.com/articles/s41467-022-31457-3) Nature Communications, 2023.
>
> [3] Bianchi,et al. [Safety-tuned llamas: Lessons from improving the safety of large language models that follow instructions](https://arxiv.org/abs/2309.07875). ICLR 2023.
>
> [4] Liu, et al. [Enhancing LLM Safety via Constrained Direct Preference Optimization.](https://arxiv.org/abs/2403.02475) ICLR WorkShop 2024.
>
> **Weakness 2: The task setting is overly simplistic. Although the structure of the antibody's FR region is relatively conserved and can be considered known, the binding pose between the antibody and antigen is typically unknown. However, given that the main goal of this work is to propose a new method for antibody optimization, this limitation is understandable.**
>
> Thank you for your insightful comment regarding the limitation of our method. When the binding pose between the antibody and antigen is unknown, recent approaches in antibody design [1] utilize protein structure prediction and docking software to determine the pose before designing antibodies. In response to your concern, we conducted a new experiment where we first establish the binding pose following the strategy used in these methods, and then design the CDR regions with AbNovo.
>
> As demonstrated in the table below, AbNovo continues to outperform other methods in this application scenario. We have included this table in Appendix (Table 9, Line 886).
>
> |  | AAR H3 | RMSD H3 | Rosetta Binding Energy | Evolutionary Plausibility | Constraints |
> | --- | --- | --- | --- | --- | --- |
> | dyMEAN | 0.37 | 3.88 | -1.75 | 2.82 | 94.5% |
> | AbX | 0.40 | 2.83 | 11.22 | **2.54** | 39.9% |
> | AbNovo | **0.44** | **2.59** | **-5.81** | **2.54** | **25.5%** |
>
> [1] Luo, et al. [Antigen-Specific Antibody Design and Optimization with Diffusion-Based Generative Models for Protein Structures.](https://openreview.net/pdf?id=jSorGn2Tjg) NeurIPS 2022.

---

> ### Author Response · Authors · 2024-11-21
>
> **Question 1: The announcement of "The first deep generative model for multi-objective antibody design" in summarized contributions, AbDPO also supports multi-objective optimization.**
>
> Thank you for your suggestion. We have removed the word "first" from that sentence (Line 71).
>
>
>
> **Question 2: Question about energy minimization.**
>
> In our initial manuscript, following the post-processing procedures for designed antibodies in previous method [1, 2], we conducted the energy minimization for both backbone and side-chain structures.
>
> In response to your concern,  we have added a new experiment where we relaxed only side-chain atoms while keeping all backbone atoms fixed. As demonstrated in the table below, AbNovo continues to outperform other methods in this setting (row “w.o. fixed backbone” and row “fixed backbone”).
>
> |  | dyMEAN | AbX | DiffAb | AbNovo |
> | --- | --- | --- | --- | --- |
> | w.o. fixed backbone | -1.7 | 4.8 | -1.0 | **-12.1** |
> | fixed backbone | 607.5 | 457.1 | 427.3 | **89.4** |
> | fixed backbone w.o. rep | -7.2 | -9.8 | -6.6 | **-17.9** |
>
> We can also observe that  the energy regarding to all methods increased significantly. This may suggest that the designed backbone atoms  have steric clashes that cannot be resolved without optimizing the backbone conformation. To validate this, we recalculated the Rosetta energy by removing the *atomic repulsion energy* term related to steric clashes. The energy scores drop substantially, suggesting that steric clashes in the fixed backbone contribute to the high energies observed (the third row named fixed backbone w.o. rep).
>
> Regarding your question about whether these experiments demonstrate that AbNovo simply provides a better initial structure for Rosetta relaxation, we also computed the backbone structural deviations between pre- and post-relaxation in the setting where both backbone and side chain are relaxed.  As demonstrated in the table below, since the backbone structures change very little during relaxation (less than 0.1 Å), this indicates that AbNovo inherently generates high-quality backbone conformations that do not heavily rely on relaxation for improvement.
>
> |  | dyMEAN | AbX | DiffAb | AbNovo |
> | --- | --- | --- | --- | --- |
> | RMSD between relaxed and unrelaxed | 0.07 | 0.10 | 0.09 | 0.07 |
>
> [1] Luo, et al. [Antigen-Specific Antibody Design and Optimization with Diffusion-Based Generative Models for Protein Structures.](https://openreview.net/pdf?id=jSorGn2Tjg) NeurIPS 2022.
>
> [2] Zhu, et al. [Antibody Design Using a Score-based Diffusion Model Guided by Evolutionary.](https://openreview.net/pdf?id=1YsQI04KaN) ICML 2024.
>
> **Question 3: Does the optimization of these physical properties contribute to some chemical validity? For example, does the peptide bond length get closer to the actual length?**
>
> We found that through optimization of these physical properties, although steric clashes cannot be completely avoided, they can be further minimized to improve chemical validity.
> Here, we evaluate the chemical validate with the mean absolute  error (MAE) of peptide bond length between native antibodies and designed antibodies. When comparing our base model and the preference-optimized model, we found that after preference optimization, the peptide bond length became closer to the actual length.
> Moreover, AbNovo outperformed other comparative methods in terms of peptide bond length.
>
> |  | dyMEAN | DiffAb | AbX | AbNovo base | AbNovo |
> | --- | --- | --- | --- | --- | --- |
> | peptide bond length MAE | 0.46 | 0.56 | 0.31 | 0.35 | **0.24** |
>
> **Question 4: The standard deviation of the physical energy needs to be presented.**
>
> Following you suggestion, we have now computed the standard deviation of the physical energy. Specifically, for each designed antibody, we ran Rosetta 10 times and report the standard deviations in the table below. The low standard deviation across all methods indicates that this benchmarking metric is robust.
>
> | dyMEAN | AbX | DiffAb | AbNovo |
> | --- | --- | --- | --- |
> | 1.2 | 2.3 | 1.0 | **0.9** |
>
> **Question 5: The AAR performance is excessively high, and it's necessary to check whether the training data of the protein language model contains samples similar to the test set.**
>
> The structure-aware language model used by AbNovo is trained on PDB and AFDB structural data. When training this language model, we strictly filtered out all antibody structures and sequences from the training dataset.
>
> **Question 6: I am curious about how many amino acids have mutated in those designed antibodies that outperform natural ones (at least in binding energy).**
>
> Following your suggestion, we analyzed the designed antibodies that outperform the natural ones in terms of binding energy. On average, 44.9% (standard deviation of 5.8%) of amino acids were mutated in the CDR H3 region, and 19.8% (standard deviation of 2.5%) of amino acids were mutated across all CDR regions.

---

> > ### Author Response · Authors · 2024-11-21
> >
> > **Question 7: The task setting of dyMEAN is different from others, including AbNovo. dyMEAN does not provide the real FR structure, making direct comparison somewhat unfair. Additionally, how is it achieved to use dyMEAN to generate 128 antibodies for an antigen?**
> >
> > We agree that differences in task settings for dyMean can lead to unfair evaluations.  We include dyMEAN in our comparison because it is a widely used method and a strong baseline in the literature. We have now explicitly stated that dyMEAN's experimental setup and the potential biases it may introduce in the main text (Line 391-394).
> >
> > Additionally, we have clarified that dyMEAN does not generate diverse samples, and we only sample 128 antibodies for all generative models. In our previous submission, we did not show the distribution of designed antibodies for a specific antigen for dyMEAN due to the same reason (Figure 3, Line 920).
> >
> > **Question 8: Question about structure alignment.**
> >
> > In our initial submission, we applied structural alignment for  all methods when calculating RMSD metric.
> >
> > In response to your concern,  we have recomputed the RMSD metric without performing a structural alignment.  As shown in the table below, the RMSD values with and without structural alignment exhibited  only a very slight deviation (less than 0.005 Å). This small difference arises because the framework region dominates the majority of the antibody Fv region. AbNovo still outperforms all baseline methods.
> >
> > |  | DIffAb | dyMEAN | AbX | GeoAb | AbNovo |
> > | --- | --- | --- | --- | --- | --- |
> > | RMSD with structural alignment | 2.86 | 3.88 | 2.49 | 2.57 | 2.37 |
> > | RMSD without structural alignment | 2.86 | 3.88 | 2.50 | 2.57 | 2.38 |
> >
> > We have now updated the result tables (Table 5, Line 810).
> >
> > Thank you for your constructive feedback on our paper. If you have any further concerns, please feel free to discuss them with us. We look forward to your response.

---

> > > ### Comment · Reviewer_vnHk · 2024-11-24
> > >
> > > I’m satisfied with the response and has increased my score to 6. Well done

---

> > > > ### Author Response · Authors · 2024-11-24
> > > >
> > > > We sincerely appreciate the reviewer's constructive comments again, which made the quality of our manuscript improved greatly.

---

### Official Review · Reviewer_neD7 · 2024-11-04

**Soundness:** 2
**Presentation:** 3
**Contribution:** 3
**Rating:** 6
**Confidence:** 3

**Summary:**

This paper focuses on some important properties, such as non-antigen binding specificity and low self-association, and optimizes the model in a DPO-like manner. What differs it from other DPO-based methods lies in two forms, the optimization targets and continuous rewards. With a two stages training framework, the proposed AbNovo is capable of capturing generalized protein information and constraining the generated results with desired properties. Experiments also support the effectiveness that generated antibodies are well designed.

**Strengths:**

1. Multiple objects are considered to improve the quality of generated antibodies. Although not validated in wet lab, these kinds of properties are essential.
2. This work does not simply integrate DPO only optimizing binding affinity, which broadens the horizons for similar works.

**Weaknesses:**

1. Rosetta energy is used as an alignment metric. It is well-known that forcefield energies have a weak correlation with measured binding affinity, typically around 0.3 [1,2]. This may lead to the totally wrong direction.
2. Limited antibody optimization experiments, which should be a major highlight of antibody design. Maybe some further experiment may alleviate this, like in [3,4].

[1]Luo S, Su Y, Wu Z, et al. Rotamer density estimator is an unsupervised learner of the effect of mutations on protein-protein interaction[J]. bioRxiv, 2023: 2023.02. 28.530137.

[2]Ambrosetti, F., Piallini, G., & Zhou, C. Evaluating Forcefield Energies in Protein Binding Studies. National Center for Biotechnology Information, 2020.

[3]Kong X, Huang W, Liu Y. End-to-end full-atom antibody design[J]. arXiv preprint arXiv:2302.00203, 2023.

[4]Shitong Luo, Yufeng Su, Xingang Peng, Sheng Wang, Jian Peng, and Jianzhu Ma. Antigen-specific
antibody design and optimization with diffusion-based generative models for protein structures.
Advances in Neural Information Processing Systems, 35:9754–9767, 2022.

**Questions:**

1. In the visualization part, I don't see why results come from dyMEAN and DiffAb do not satisfy constraints like Stability, Self-association. Can you explain this in detail?

---

> ### Author Response · Authors · 2024-11-21
>
> Thank you for valuable suggestions, which have helped us significantly improve the quality of our manuscript. We have provided detailed responses point-to-point to your comments as follows. We hope that our responses and additional experiments address your concerns.
>
> **Weakness 1 : Rosetta energy is used as an alignment metric. It is well-known that forcefield energies have a weak correlation with measured binding affinity, typically around 0.3 [1,2]. This may lead to the totally wrong direction.**
>
> Thank you for raising this important concern. We agree that forcefield energies are limited in measuring binding affinities.  We would like to clarify our methods and the rationale for using forcefield energies as follows.
>
> First, our method does not rely solely on forcefield energies for optimization. Instead, we employ a unified framework of constrained optimization that integrates multiple objectives and constraints to guide the optimization process and prevent it from diverging in an incorrect direction. Specifically,
>
> 1. We include the likelihood under a large-scale protein language model as an optimization objective, which has proven effective in improving antibody screening success rates in wet-lab experiments [1, 2].
> 2.  We incorporate constraints for other biophysical properties such as specificity and low self-association, which have also proven useful in guiding antibody design in experimental settings [3, 4].
> 3. During preference optimization, we introduce a regularization term in the training loss function (Equation 3, Line 236, Page 5) to ensure the fine-tuned model remains close to the base model, preventing arbitrary divergence.
>
> Second, despite its limitations, Rosetta binding energy has been utilized as a screening metric, and several proteins designed using it have been experimentally validated in wet-lab experiments [5, 6]. Thus, binding energy is widely used as a metric for benchmarking recent generative models for antibody design, such as DiffAb [7], dyMEAN [8], AbX [9], and  AbDPO [10].
>
> Third,  the AbNovo framework is flexible and allows the inclusion of other constraints or optimization objectives important for antibody design.  One of our key methodological contributions lies in building the framework of constrained preference optimization for antibody design, providing rigorous theoretical derivations and proofs.
>
> Finally, we acknowledge that, although forcefield energy is widely used as a metric in recent work, there remains a gap in its correlation with experimental results. We have included this point in the Discussion section (Line 533) to raise awareness in the community and inspire the development of improved methods.
>
> [1] Hie, et al. [Efficient evolution of human antibodies from general protein language models.](https://www.nature.com/articles/s41587-023-01763-2) Nature Biotechnology 2024.
>
> [2] Shuai, et al. [IgLM: Infilling language modeling for antibody sequence design](https://www.cell.com/cell-systems/fulltext/S2405-4712(23)00271-5). Cell System 2023.
>
> [3]  Makowski, et al. [Optimization of therapeutic antibodies for reduced self-association and non-specific binding via interpretable machine learning.](https://www.nature.com/articles/s41551-023-01074-6) Nature Biomedical Engineering 2023.
>
> [4] Makowski, et al. [Co-optimization of therapeutic antibody affinity and specificity using machine learning models that generalize to novel mutational space.](https://www.nature.com/articles/s41467-022-31457-3) Nature Communications, 2023.
>
> [5] Cao, et al. [Design of protein-binding proteins from the target structure alone.](https://www.nature.com/articles/s41586-022-04654-9) Nature 2022.
>
> [6] Sun, et al. [Accurate de novo design of heterochiral protein–protein interactions.](https://www.nature.com/articles/s41422-024-01014-2#Sec9)  Cell Research 2024.
>
> [7] Luo, et al. [Antigen-Specific Antibody Design and Optimization with Diffusion-Based Generative Models for Protein Structures.](https://openreview.net/forum?id=jSorGn2Tjg) NeurIPS 2022.
>
> [8] Kong, et al. [End-to-End Full-Atom Antibody Design](https://arxiv.org/abs/2302.00203). ICML 2023.
>
> [9] Zhu, et al. [Antibody Design Using a Score-based Diffusion Model Guided by Evolutionary.](https://openreview.net/pdf?id=1YsQI04KaN) ICML 2024.
>
> [10] Zhou, et al. [Antigen-Specific Antibody Design via Direct Energy-based Preference Optimization.](https://arxiv.org/pdf/2403.16576v1) NeurIPS 2024.

---

> > ### Author Response · Authors · 2024-11-21
> >
> > **Weakness 2: Limited antibody optimization experiments, which should be a major highlight of antibody design. Maybe some further experiment may alleviate this, like in [3,4].**
> >
> > Following your suggestion, we have added new experiments on antibody optimization.  Specifically,  we adopt the protocol and methods used in DiffAb for antibody optimization and  we show the performance of different methods under varying optimization steps. This process involves perturbing the CDR sequence and structure at time $t$ using forward diffusion, then denoising from the time $t$ to time $0$ in reverse diffusion to generate 128 antibodies for each antigen.
> >
> > As shown in the following table, AbNovo has better performance in Rosetta Binding Energy, Evolutionary Plausibility, AAR, RMSD, and the Proportion of constraint satisfaction.
> >
> > We have added these new results to our manuscript (Appendix Table 6, Table 7, and Table 8) (from line 838 to line 873).
> >
> > | Optimization steps | DiffAb(Rosetta Binding Energy /Evolutionary Plausibility | AbX (Rosetta Binding Energy /Evolutionary Plausibility) | AbNovo  (Rosetta Binding Energy /Evolutionary Plausibility) |
> > | --- | --- | --- | --- |
> > | 4 | -10.45/2.39 | -8.80/2.40 | **-21.02/2.39** |
> > | 8 | -8.52/2.41 | -2.64/2.43 | **-19.77/2.37** |
> > | 16 | -7.18/2.42 | 2.07/2.42 | **-12.70/2.37** |
> > | 32 | -6.50/2.53 | -3.50/2.44 | **-15.35/2.36** |
> > | 64 | 0.23/2.57 | 3.98/2.44 | **-12.87/2.36** |
> > | 100 | -0.96/2.60 | 4.79/2.44 | **-12.05/2.36** |
> >
> > | Optimization steps | DiffAb (Constraints) | AbX (Constraints) | AbNovo (Constraints) |
> > | --- | --- | --- | --- |
> > | 4 | 13.2% | 14.0% | **12.8%** |
> > | 8 | 13.9% | 22.7% | **7.1%** |
> > | 16 | 13.6% | 22.5% | **6.5%** |
> > | 32 | 15.7% | 21.9% | **4.2%** |
> > | 64 | 21.5% | 23.0% | **3.6%** |
> > | 100 | 20.8% | 23.5% | **3.9%** |
> >
> > | Optimization steps | DiffAb (AAR / RMSD) | AbX (AAR / RMSD) | AbNovo (AAR / RMSD) |
> > | --- | --- | --- | --- |
> > | 4 | **0.88** / 1.09 | 0.80 / 0.97 | 0.85 **/ 0.80** |
> > | 8 | **0.76** / 1.59 | 0.59 / 1.51 | 0.69 / **1.34**  |
> > | 16 | 0.48 / 1.78 | 0.49 / 1.54 | **0.51 / 1.46** |
> > | 32 | 0.39 / 2.05 | 0.45 / 1.88 | **0.50 / 1.66** |
> > | 64 | 0.30 / 2.69 | 0.45 / 2.33 | **0.48 / 2.03** |
> > | 100 | 0.28 / 2.86 | 0.44 / 2.50 | **0.49 / 2.38** |
> >
> >
> > **Queation 1: In the visualization part, I don't see why results come from dyMEAN and DiffAb do not satisfy constraints like Stability, Self-association. Can you explain this in detail?**
> >
> > Following your suggestion, we have revised Figure3 to demonstrate more details.  Specifically, we presented the sequence on CDR H3 region  annotated with biochemical properties. We explain the details for this case regarding Stability and Self-Association as follows:
> >
> > **Stability:**
> >
> > *Stability* measures the stability of the conformation of designed antibody in isolation, without the antigen structure involved. Following the protocol in previous method, we compute the metric of Stability using Rosetta Software.
> > We observed that the structures generated by dyMEAN exhibit numerous steric clashes (indicated by dashed lines in the structure), which will break down the energy term of  van der Waals force in Rosetta Energy.
> >
> > **Self-association**
> >
> > *Self-Association* refers to the tendency of antibody molecules to aggregate with each other.
> > Previous studies have shown that a larger area of negatively charged patches in the CDRs corresponds to a higher risk of self-association in wet-lab experiments [1]. We see that dyMEAN produce a large number of charged amino acids which can lead to potential risks of self-association.
> >
> > Please note that the case of DiffAb used in this version of the manuscript differs from the previous version, following the suggestion of reviewer fXjU. In this updated evaluation, the antibodies designed by DiffAb satisfy the Stability and Self-association constraints. However, they exhibit poor performance in other critical metrics, including Rosetta binding energy, Evolutionary Plausibility, AAR, and RMSD.
> >
> > [1] Makowski, et al. [Optimization of therapeutic antibodies for reduced self-association and non-specific binding via interpretable machine learning](https://www.nature.com/articles/s41551-023-01074-6). Nature Biomedical Engineering 2023.
> >
> >
> > Thank you for your constructive feedback on our paper. If you have any further concerns, please feel free to discuss them with us. We look forward to your response.

---

> > > ### Comment · Reviewer_neD7 · 2024-11-23
> > >
> > > Thanks for the response, and I have raised my score.

---

> > > > ### Author Response · Authors · 2024-11-24
> > > >
> > > > We sincerely appreciate the reviewer's constructive comments again, which made the quality of our manuscript improved.

---

### Public Comment · ~Kiwoong_Yoo1 · 2025-09-01
**Code repo gone**

Dear authors, the code repo seems to be invalid, can you fix this issue? Thanks

---

### Meta-Review · Area_Chair_wr12 · 2024-12-20

**Metareview:**

This paper introduces a multi-objective antibody design method, AbNovo. The authors train an antigen conditioned generative model for antibodies, and then fine tune to maximize binding affinity subject to constraints on biophysical properties like high stability and low self-association. The authors achieve stronger performance on the RAbD in silico antibody design test set than many recent methods in this area. Although the authors' method seems to differ from AbDPO primarily in the introduction of constraints into the preference optimization, the final empirical results are quite strong relative to prior work.

**Additional Comments On Reviewer Discussion:**

The authors addressed with new experimental data two major concerns raised by the reviewers: (1) the relative lack of antibody optimization results in the original paper, and (2) optimization results when the antibody-antigen binding pose is not known in advance. Beyond these, the authors also cleaned up the paper substantially, with several missing technical terms and metric definitions now provided in the current draft of the paper. Looking through the draft, these updates (e.g., the inclusion of the new Appendix A.5 defining evaluation metrics, and additional text throughout the message section) appear adequate. The reviewers are in agreement, and after a score raise are unanimously in favor of acceptance.

---

### Decision · Program_Chairs · 2025-01-22

Accept (Poster)